# Rethinking Expressivity and Degradation-Awareness in Attention for All-in-One Blind Image Restoration

**Bin Ren[1,2]\*   Runyi Yang[3]\*   Qi Ma[4,3]   Xu Zheng[5,3]   Mengyuan Liu[6,7]†   Danda Pani Paudel[3]**
**Luc Van Gool[3]   Rita Cucchiara[8]   Nicu Sebe[2]**

[1]Mohamed bin Zayed University of Artificial Intelligence   [2]University of Trento   [3]INSAIT, Sofia University "St. Kliment Ohridski"   [4]ETH Zürich

[5]HKUST (GZ)   [6]State Key Laboratory of General Artificial Intelligence, Peking University, Shenzhen Graduate School

[7]Guangdong Provincial Key Laboratory of Ultra High Definition Immersive Media Technology, Peking University, Shenzhen Graduate School

[8]University of Modena and Reggio Emilia

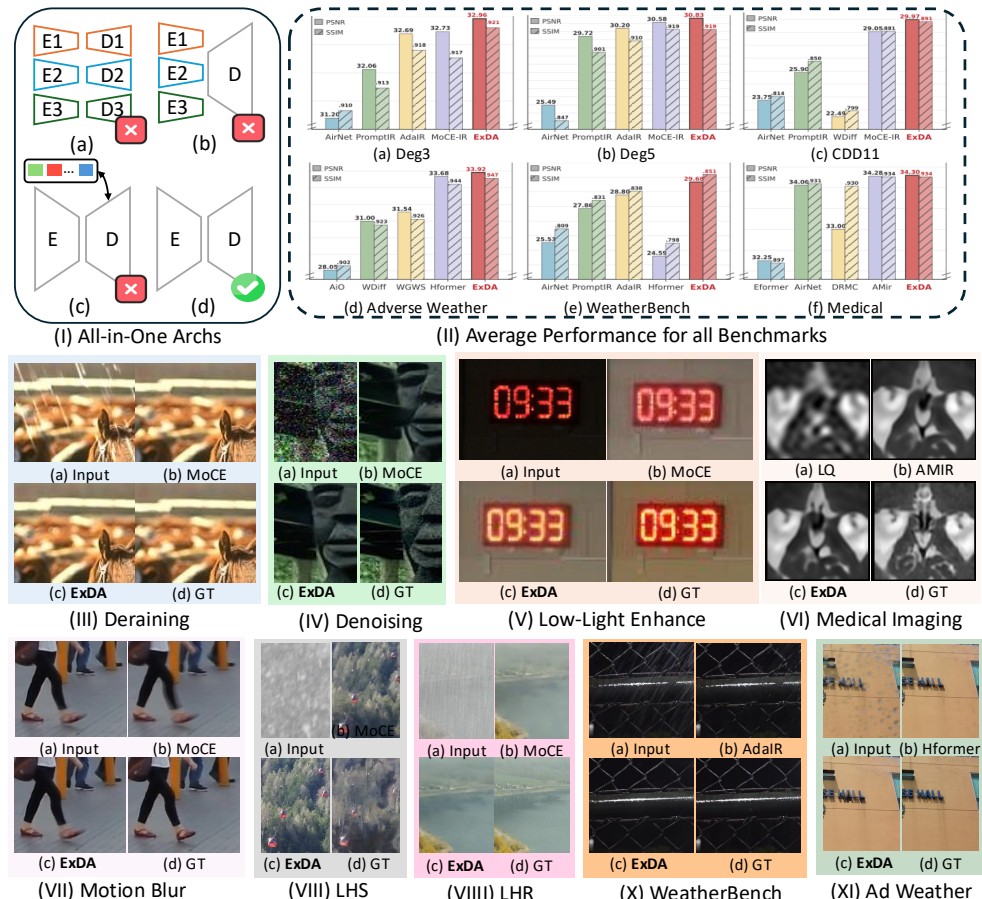

Figure 1: *(I) Architectures:* We propose that the simple encoder-decoder (d) pipeline is strong enough for All-in-One IR. *(II) Average Performance:* our ExDA consistently produced better results across six challenging All-in-One benchmarks. *(III–XI) Qualitative Comparison:* Please zoom in for details.

## Abstract

All-in-one image restoration (IR) aims to recover high-quality images from diverse degradations, which in real-world settings are often mixed and unknown. Unlike single-task IR, this problem requires a model to approximate a family of heterogeneous inverse functions, making it fundamentally more challenging and practically important. Although recent focus has shifted toward large multimodal models, their robustness still depends on faithful low-level inputs, and the principles that govern effective restoration remain underexplored. We revisit attention mechanisms through the lens of all-in-one IR and identify two overlooked bottle-

---

\* indicates equal contribution; † indicates corresponding author: Mengyuan Liu <liumengyuan@pku.edu.cn>.

necks in widely adopted Restormer-style backbones: *(i) the value path remains purely linear*, restricting outputs to the span of inputs and weakening expressivity, and *(ii) the absence of an explicit global slot* prevents attention from encoding degradation context. To address these issues, we propose two minimal, backbone-agnostic primitives: a nonlinear value transform that upgrades attention from a selector to a selector–transformer, and a global spatial token that provides an explicit degradation-aware slot. Together, these additions improve restoration across synthetic, mixed, underwater, and medical benchmarks, with negligible overhead and consistent performance gains. Analyses with foundation model embeddings, spectral statistics, and separability measures further clarify their roles, positioning our study as a step toward rethinking attention primitives for robust all-in-one IR.

# 1 INTRODUCTION

Image restoration (*i.e.*, IR) aims to recover high-quality images from degraded observations. In real-world scenarios, images often suffer from complex degradations—noise, blur, haze, rain, and their unpredictable mixtures—each corresponding to a distinct inverse function that must be learned and applied adaptively. This multiplicity of inverse mappings makes all-in-one image restoration fundamentally more challenging than single-task variants: rather than learning one specialized function, the model must approximate a *family of heterogeneous inverse transformations* while remaining robust to mixed and unseen degradations. The stakes for robust low-level vision have never been higher. As computer vision increasingly relies on large multimodal models and foundation architectures, their effectiveness hinges critically on the quality of input images. Corrupted inputs propagate errors throughout the entire pipeline, making reliable image restoration not merely a specialized tool, but a *foundational prerequisite* for robust vision systems.

Recent advances are dominated by Transformer-based architectures that skillfully balance global context modeling with local spatial priors. The multi-dconv head transposed attention (MDTA) and gated-dconv feed-forward network (GDFN) introduced in Restormer (Zamir et al., 2022) exemplify this trend, replacing computationally prohibitive token-wise self-attention with efficient *channel-wise attention*. This design has become the de-facto standard for high-resolution IR, spawning numerous variants and establishing strong empirical baselines across diverse restoration tasks. However, when viewed through the lens of all-in-one IR, this prevalent design exposes two critical yet overlooked limitations that reflect a deeper tension between single-task efficiency and unified IR capability.

First, the attention mechanism's value path remains purely linear: while queries and keys interact through nonlinear softmax operations, values are merely linearly aggregated, *constraining outputs to lie within the convex hull of input features*. This expressivity bottleneck becomes particularly severe in all-in-one settings, where the model must navigate between vastly different inverse mappings—from high-frequency noise removal to low-frequency haze correction—yet is constrained by linear combinations of its inputs. Although (Katharopoulos et al., 2020; Shen et al., 2024; Aksenov et al., 2024; Shazeer, 2020) use of nonlinearities on queries and keys enables an efficient linearized softmax, we argue that the value space is more critical for learning robust representations. Second, unlike standard ViTs that employ class tokens for global aggregation, channel-wise attention *discards the notion of explicit global slots entirely*. This forces the degradation context to be encoded implicitly across spatial channels, making the model less capable of explicit degradation inference—a capability that proves essential when corruption types are unknown and mixed. These limitations matter less in single-task settings where the inverse function is fixed and known, but become fundamental bottlenecks in all-in-one scenarios that demand both expressivity and adaptability.

In this work, we address these bottlenecks through a principled rethinking of attention primitives for all-in-one IR. We introduce two minimal, backbone-agnostic extensions that transform any Restormer-style architecture into a more expressive and degradation-aware system. Our **nonlinear value transform** breaks the linear span constraint by augmenting values with lightweight convolutional mappings *before* aggregation, upgrading attention from a feature selector to a feature *selector–transformer*. Our **global spatial token (GST)** restores explicit global context by providing learnable slots that adaptively aggregate degradation-relevant statistics and inject them back into the attention computation. These innovations are both theoretically motivated and empirically validated: from a function approximation perspective, we show that pre-aggregation nonlinear transformations

fundamentally expand the realizable function family, while diagnostic analyses involving foundation model embeddings, spectral decompositions, and cluster separability measures demonstrate that global tokens capture meaningful degradation context. Extensive experiments across synthetic, mixed, underwater, and medical benchmarks reveal consistent improvements over strong baselines, with negligible computational overhead and robust transfer across different backbone architectures.

Our contributions address a fundamental gap in understanding attention mechanisms for unified IR:

- We identify and theoretically analyze two overlooked bottlenecks in prevalent Restormer-style architectures—linear value constraints that limit expressivity and absent global slots that limit degradation-awareness.
- We propose two minimal, backbone-agnostic primitives that address these bottlenecks: nonlinear value transforms that expand the realizable function family, and global spatial tokens that provide explicit degradation context.
- We provide theoretical motivation, diagnostic analyses, and extensive empirical validation across diverse benchmarks, demonstrating consistent improvements while maintaining computational efficiency.

## 2 RELATED WORK

**Image Restoration (IR).** IR aims to solve a highly ill-posed problem: reconstructing high-quality images from degraded observations. Owing to its broad importance, IR has been widely applied in numerous applications (Richardson, 1972; Xie et al., 2025; Banham & Katsaggelos, 1997; Li et al., 2023b; Zamfir et al., 2024). Early approaches were dominated by model-based solutions that searched for closed-form results under handcrafted formulations. With the advent of deep learning, learning-based IR methods have rapidly gained popularity. Recent studies include regression-based techniques (Lim et al., 2017; Lai et al., 2017; Liang et al., 2021; Chen et al., 2021; Li et al., 2023a; Zhang et al., 2024) and generative pipelines (Gao et al., 2023; Wang et al., 2023b; Luo et al., 2023; Yue et al., 2023; Zhao et al., 2024), built upon a variety of architectures such as convolutional networks (Dong et al., 2015; Zhang et al., 2017b;a; Wang et al., 2018), MLP-based designs (Tu et al., 2022), state space models (Guo et al., 2024a; Zhu et al., 2024; Gu & Dao, 2023; Dao & Gu, 2024), and Vision Transformers (ViTs) (Liang et al., 2021; Ren et al., 2023a; Li et al., 2023a; Zamir et al., 2022; Dosovitskiy et al., 2020; Liu et al., 2023). Among them, Restormer (Zamir et al., 2022) introduced a channel-wise attention mechanism (MDTA) to achieve linear complexity while handling high-resolution inputs, and has since become a widely adopted backbone in IR due to its strong balance of efficiency and accuracy. Despite these advances, the majority of IR models are still designed for specific degradation types, such as denoising (Zhang et al., 2017b; 2019), dehazing (Ren et al., 2020; Wu et al., 2021), deraining (Jiang et al., 2020; Ren et al., 2019), and deblurring (Kong et al., 2023; Ren et al., 2023b).

**All-in-One Blind Image Restoration.** Training separate task-specific models for individual degradations can deliver strong results, yet maintaining one model per task is impractical and environmentally costly. In practice, images are frequently affected by mixtures of degradations, making it unrealistic to handle each corruption independently. This has motivated the study of *All-in-One image restoration*, where a single model is expected to generalize across multiple degradation types (Zamfir et al., 2025; Zeng et al., 2025; Zheng et al., 2024). Different strategies have been explored: AirNet (Li et al., 2022) learns contrastive degradation representations to guide reconstruction, whereas IDR (Zhang et al., 2023) formulates degradations as physical components and leverages a meta-learning pipeline. Prompt-based approaches (Potlapalli et al., 2024; Wang et al., 2023a; Li et al., 2023c) push this idea further by conditioning restoration on learned visual prompts, later extended to frequency-aware prompts (Cui et al., 2025) or larger-scale dynamic architectures (Dudhane et al., 2024). Although effective, these methods often incur high training costs and reduced efficiency (Cui et al., 2025), which hinders practical deployment. More recently, besides these regression-based approaches, distribution-oriented models also show decent performance while with a larger model size Tian et al. (2025); Luo et al. (2025). In this work, we take a different route. Instead of relying on auxiliary prompt modules or complex extra multi-stage strategies, we revisit the backbone design itself. In parallel, recent studies have attempted to enhance attention expressivity (Chefer et al., 2021) and incorporate degradation-aware priors directly into the backbone design (Tang et al., 2025). These directions motivate a closer examination of the attention operator itself rather than adding heavier auxiliary modules. By examining Restormer-style channel-wise attention, we identify two overlooked

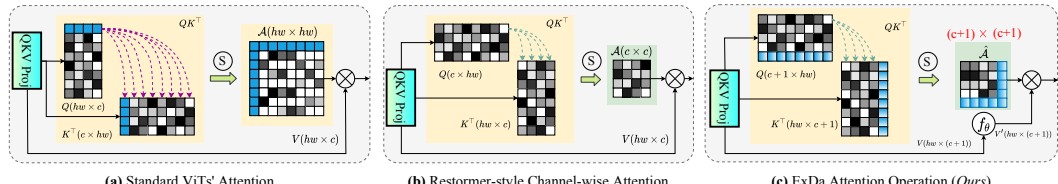

**(a)** Standard ViTs' Attention     **(b)** Restormer-style Channel-wise Attention     **(c)** ExDa Attention Operation (*Ours*)

Figure 2: Comparison of (a) standard ViT attention with CLS in high-level tasks, (b) Restormer-style channel-wise attention in IR, (c) Our proposed attention design that augments values space with nonlinearity and introduces an explicit global slot.

bottlenecks, linear values and the lack of an explicit global slot, and propose minimal primitives that directly improve expressivity and provide degradation-aware context for efficient all-in-one Blind IR.

## 3 THE PROPOSED METHOD

### 3.1 REVISITING RESTORMER-STYLE ATTENTION: HIDDEN BOTTLENECKS

Transformers have recently become central to low-level vision tasks such as image restoration (IR), where they are often re-designed to balance efficiency and spatial detail recovery. A representative example is Restormer (Zamir et al., 2022), which replaces spatial token-wise self-attention (Fig. 2a) with *channel-wise attention* (Fig. 2b) and substitutes the standard MLP feed-forward layers with gated-dconv feed-forward networks. These design choices make the model computationally feasible for high-resolution inputs and have set a strong baseline for single-degradation IR.

However, in the broader context of low-level vision, two overlooked limitations emerge. *(i)*, the *value (V) path remains purely linear*. While queries ($Q$) and keys ($K$) interact through a nonlinear softmax, values are only linearly aggregated, constraining outputs to lie within the span of the inputs. In the standard ViTs, this limitation is less severe because the MLP-based feed-forward networks provide strong nonlinear transformations. In contrast, IR backbones such as Restormer adopt gated-dconv feed-forward networks, where one branch is essentially linear. As a result, a portion of the information bypasses any nonlinear transformation, leaving the overall block with weaker nonlinearity. This makes the expressivity bottleneck of linear values particularly pronounced in *all-in-one restoration*, where the model must approximate a family of diverse inverse functions rather than a single degradation. *(ii)*, *there is no explicit global slot to summarize degradation context*. Standard ViTs include a CLS token to aggregate global semantics (Fig. 2a), yet in low-level vision, this token is often discarded as "useless" for pixel-level predictions (Liang et al., 2021; Li et al., 2023a; Ren et al., 2024). Restormer's channel-wise attention follows this practice (Fig. 2b), relying instead on local depth-wise convolutions. While sufficient for single-task restoration, this design implicitly assumes that the degradation type is fixed and known. In the all-in-one setting, however, the model must *infer* degradation type directly from the input. Without a dedicated global slot, there is no explicit mechanism to encode degradation statistics into the representation. What may appear redundant in single-task IR thus becomes indispensable in all-in-one IR, where such a slot can naturally evolve into a *degradation embedding*, capturing global statistics not only across channels but also explicitly across spatial structure (Fig. 2b). These two limitations motivate our subsequent exploration of nonlinear values and explicit global tokens.

### 3.2 NONLINEARITY MATTERS IN VALUES FOR EXPRESSIVITY

Building on the identified bottleneck in Sec. 3.1, we empirically validate the critical role of nonlinear values in all-in-one restoration. The linear value constraint severely limits the model's ability to approximate diverse inverse mappings required across different degradations.

To isolate this effect, we design experiments with multi-faceted degradations that mirror all-in-one restoration complexity. Our *synthetic function* combines nonlinear sensor response, blur kernels, additive/multiplicative noise, and quantization effects. Our *MNIST restoration* applies realistic corruptions including nonlinear response curves, motion blur, spatially-varying noise, and compression artifacts. Both scenarios require modeling multiple degradation characteristics simultaneously, analogous to all-in-one restoration challenges.

---

**Algorithm 1** GST: Content-Adaptive Global Spatial Token Generation

---

**Require:** Input features $X \in \mathbb{R}^{B \times C \times H \times W}$, heads $h$, tokens per head $K$, stride $s$
**Ensure:** Degradation-aware global tokens $G \in \mathbb{R}^{B \times h \times K \times HW}$
 1: **Efficient Spatial Compression:** $\tilde{X} = \text{AvgPool}_s(X) \in \mathbb{R}^{B \times C \times \frac{H}{s} \times \frac{W}{s}}$
 2: **Multi-Head Token Projection:** $\Phi = W_{\text{proj}} \star \tilde{X} \in \mathbb{R}^{B \times hK \times H_s \times W_s}$
 3: **Content-Adaptive Spatial Attention:**
 4: $\quad G_{\text{compact}} = \text{Softmax}_{\text{spatial}}\left(\text{Reshape}(\Phi, [B, h, K, -1])\right)$
 5: **Resolution Recovery:** $G = \text{BilinearUpsample}(G_{\text{compact}}, \text{size} = (H, W))$
 6: **return** $G \in \mathbb{R}^{B \times h \times K \times HW}$ $\qquad\qquad\qquad$ ▷ Ready for attention concatenation

---

Fig. 3 reveals consistent patterns across both settings. In the synthetic function approximation setting (a), linear value attention systematically fails in critical regions (orange highlights), achieving 50.4% worse convergence as it encounters inherent expressivity limitations. MNIST restoration (b) confirms this on realistic image data, where nonlinear values achieve 5.92 dB PSNR improvement (19.2→25.1 dB) and produce visually superior reconstructions compared to blurry, artifact-laden outputs from linear value attention.

These findings confirm our analysis: in Restormer-style architectures where gated FFN allows information to bypass nonlinear transformations, the linear value constraint becomes a critical expressivity bottleneck for all-in-one restoration requiring diverse degradation modeling. This motivates enhancing value expressivity through $V' = f_\theta(V)$, where $f_\theta$ introduces nonlinearity beyond the linear span of inputs. We adopt a residual formulation to balance preservation and transformation:

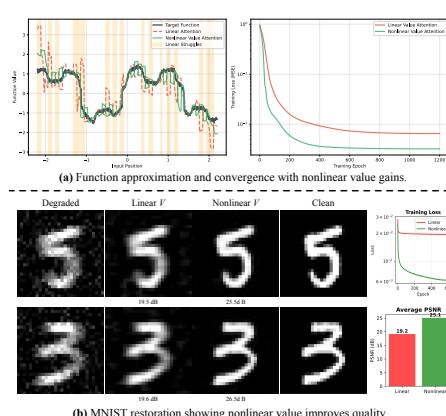

**(a)** Function approximation and convergence with nonlinear value gains.

**(b)** MNIST restoration showing nonlinear value improves quality.

Figure 3: Nonlinear value transforms outperform linear value attention in both (a) function approximation and (b) MNIST restoration, achieving better convergence, higher PSNR/SSIM, and improved quality.

$$V' = V + g_\theta(V),$$
$$g_\theta = \text{Conv}_{1\times 1} \rightarrow \text{DWConv}_{3\times 3} \rightarrow \text{GELU} \rightarrow \text{Conv}_{1\times 1}. \tag{1}$$

Two insights guide this design. First, pre-aggregation placement is essential since attention $\text{Softmax}(QK^\top/\sqrt{d})V'$ constrains outputs to linear combinations—post-aggregation nonlinearity cannot escape this fundamental limitation. Second, the lightweight $g_\theta$ provides sufficient expressivity for diverse degradation modeling while maintaining efficiency. This transforms channel-wise attention from a linear feature selector into a nonlinear feature transformer, fundamentally addressing the expressivity gap between single-task and all-in-one restoration requirements.

### 3.3 GLOBAL CONTEXT AND THE MISSING SLOT

Building on the analysis in Sec. 3.1, we now address the second identified limitation: the absence of explicit global context mechanisms in Restormer-style architectures. While Sec. 3.2 tackled the expressivity bottleneck through nonlinear value transforms, the *degradation inference* challenge requires a fundamentally different approach, introducing explicit degradation-aware slots that can capture global spatial statistics. The core issue is that without dedicated global tokens, degradation context can only be distributed implicitly across local channel interactions or captured through cross-channel attention mechanisms. This becomes particularly problematic in all-in-one settings where the model must distinguish between fundamentally different corruption types. These global spatial signatures are best captured through explicit global analysis rather than relying solely on local depth-wise convolutions or implicit cross-channel attention.

We address this through *Global Spatial Tokens (GST)*, which provide explicit degradation-aware slots while maintaining computational efficiency. As detailed in Alg. 1, our design generates content-adaptive spatial attention maps through stride-$s$ spatial compression and learnable projection. The key innovation lies in content-adaptive pooling: rather than fixed global averaging, GST learns spatial attention maps through learnable projections followed by spatial softmax normalization, enabling each

Table 1: *Comparison to state-of-the-art on **three degradations***. PSNR (dB, ↑) and SSIM (↑) metrics are reported on the full RGB images. **Best** performances is highlighted. '-' means unreported results.

| Method | Venue. | Params. | Dehazing | | Deraining | | Denoising | | | | | | Average | |
|---|---|---|---|---|---|---|---|---|---|---|---|---|---|---|
| | | | SOTS | | Rain100L | | BSD68$_{\sigma=15}$ | | BSD68$_{\sigma=25}$ | | BSD68$_{\sigma=50}$ | | | |
| BRDNet (Tian et al., 2020) | NN'20 | - | 23.23 | .895 | 27.42 | .895 | 32.26 | .898 | 29.76 | .836 | 26.34 | .693 | 27.80 | .843 |
| LPNet (Gao et al., 2019) | CVPR'19 | - | 20.84 | .828 | 24.88 | .784 | 26.47 | .778 | 24.77 | .748 | 21.26 | .552 | 23.64 | .738 |
| FDGAN (Dong et al., 2020) | AAAI'20 | - | 24.71 | .929 | 29.89 | .933 | 30.25 | .910 | 28.81 | .868 | 26.43 | .776 | 28.02 | .883 |
| DL (Fan et al., 2019) | TPAMI'19 | 2M | 26.92 | .931 | 32.62 | .931 | 33.05 | .914 | 30.41 | .861 | 26.90 | .740 | 29.98 | .876 |
| MPRNet (Zamir et al., 2021) | CVPR'21 | 16M | 25.28 | .955 | 33.57 | .954 | 33.54 | .927 | 30.89 | .880 | 27.56 | .779 | 30.17 | .899 |
| AirNet (Li et al., 2022) | CVPR'22 | 9M | 27.94 | .962 | 34.90 | .967 | 33.92 | .933 | 31.26 | .888 | 28.00 | .797 | 31.20 | .910 |
| NDR (Yao et al., 2024) | TIP'24 | 28M | 25.01 | .860 | 28.62 | .848 | 28.72 | .826 | 27.88 | .798 | 26.18 | .720 | 25.01 | .810 |
| PromptIR (Potlapalli et al., 2024) | NeurIPS'23 | 36M | 30.58 | .974 | 36.37 | .972 | 33.98 | .933 | 31.31 | .888 | 28.06 | .799 | 32.06 | .913 |
| MoCE-IR-S (Zamfir et al., 2025) | CVPR'25 | 11M | 30.98 | .979 | 38.22 | .983 | 34.08 | .933 | 31.42 | .888 | 28.16 | .798 | 32.57 | .916 |
| AdaIR (Cui et al., 2025) | ICLR'25 | 29M | 31.06 | .980 | 38.64 | .983 | 34.12 | .935 | 31.45 | .892 | 28.19 | .802 | 32.69 | .918 |
| MoCE-IR (Zamfir et al., 2025) | CVPR'25 | 25M | 31.34 | .979 | 38.57 | .984 | 34.11 | .932 | 31.45 | .888 | 28.18 | .800 | 32.73 | .917 |
| ExDA *(Ours)* | ICLR'26 | 22M | **31.58** | **.982** | **39.13** | **.985** | **34.22** | **.937** | **31.56** | **.895** | **28.32** | **.808** | **32.96** | **.921** |
| Methods with the assistance of vision language, multi-task learning, natural language prompts, or multi-modal control | | | | | | | | | | | | | | |
| DA-CLIP (Luo et al., 2024) | ICLR'24 | 125M | 29.46 | .963 | 36.28 | .968 | 30.02 | .821 | 24.86 | .585 | 22.29 | .476 | - | - |
| Art$_{PromptIR}$ (Wu et al., 2024) | ACM MM'24 | 36M | 30.83 | .979 | 37.94 | .982 | 34.06 | .934 | 31.42 | .891 | 28.14 | .801 | 32.49 | .917 |
| InstructIR-3D (Conde et al., 2024) | ECCV'24 | 16M | 30.22 | .959 | 37.98 | .978 | 34.15 | .933 | 31.52 | .890 | 28.30 | .804 | 32.43 | .913 |
| UniProcessor (Duan et al., 2025) | ECCV'24 | 1002M | 31.66 | .979 | 38.17 | .982 | 34.08 | .935 | 31.42 | .891 | 28.17 | .803 | 32.70 | .918 |
| VLU-Net (Zeng et al., 2025) | CVPR'25 | 35M | 30.71 | .980 | 38.93 | .984 | 34.13 | .935 | 31.48 | .892 | 28.23 | .804 | 32.70 | .919 |
| Perceive-IR (Zhang et al., 2025) | TIP'25 | 42M | 30.87 | .975 | 38.29 | .980 | 34.13 | .934 | 31.53 | .890 | 28.31 | .804 | 32.63 | .917 |
| DFPIR (Tian et al., 2025) | CVPR'25 | 31M | 31.87 | .980 | 38.65 | .982 | 34.12 | .935 | 31.47 | .893 | 28.25 | .806 | 32.88 | .919 |
| DA-RCOT (Tang et al., 2025) | TPAMI'25 | 50M | 31.26 | .977 | 38.36 | .983 | 33.98 | .934 | 31.33 | .890 | 28.10 | .801 | 32.60 | .917 |

token to naturally develop distinct spatial emphasis patterns that correspond to different degradation characteristics. This enables natural specialization during training—noise tokens activate scattered high-frequency regions, blur tokens emphasize smooth low-frequency areas, and haze tokens respond to large-scale illumination structures, all without explicit supervision or degradation labels.

The generated GST tokens $G \in \mathbb{R}^{B \times h \times K \times HW}$ are seamlessly integrated into channel-wise attention through direct concatenation with query, key, and value representations:

$$[Q^*, K^*, V^*] = [Q \oplus G, K \oplus G, V' \oplus G] \in \mathbb{R}^{B \times h \times (C+K) \times HW}, \quad (2)$$

where $\oplus$ denotes channel-wise concatenation and $V'$ is the nonlinear-enhanced value from Sec. 3.2. After standard attention computation $\text{Attn} = \text{Softmax}(Q^* K^{*\top}/\tau)V^*$, we separate local channel features and global token contributions, re-injecting GST influence via learnable residual scaling:

$$\text{Output} = \text{Attn}_{[:,:,:C,:]} + \alpha \cdot \text{Attn}_{[:,:,C:,:]}, \quad (3)$$

where $\alpha$ is initialized to 0.1 for gradual learning without overwhelming local features.

This design transforms the discarded CLS token concept into a valuable primitive for all-in-one scenarios. By providing explicit degradation-aware slots that naturally specialize during training, GST enables coherent global strategies while maintaining channel-wise attention efficiency. The content-adaptive nature allows each attention head to learn specialized global patterns without requiring explicit degradation labels or supervisory signals.

## 4 EXPERIMENTS

We evaluate ExDA under **six** All-in-One IR benchmarks, *i.e.*, *(i) All-in-One (3Degradations)*, *(ii) All-in-One (5Degradations)*, *(iii) Mixed Degradation*, *(iv) Adverse Weather Removal*, *(v) Real-world WeatherBench*, and *(vi) Medical All-in-One*. The macro architecture of our model, the datasets, and implementation details are provided in our appendix. Project page:https://amazingren.github.io/ExDA/.

### 4.1 MAIN RESULTS.

**3-Degradation Setting.** The results in Tab. 1 show that ExDA consistently outperforms other methods, even outperforming those with the assistance of language, multi-task, or text/visual prompts. Notably, 0.23 dB higher than the recent MoCE-IR on PSNR, while with 3M fewer parameters.

**5-Degradation Setting.** Building on the previous setting, we further include deblurring and low-light enhancement as two additional degradation types (Li et al., 2022; Zhang et al., 2023). As shown in

Table 2: *Comparison to state-of-the-art on **five degradations***. PSNR (dB, ↑) and SSIM (↑) metrics are reported on the full RGB images with (*) denoting general image restorers, others are specialized all-in-one approaches. **Best** performance is highlighted.

| Method | Venue | Params. | Dehazing SOTS | | Deraining Rain100L | | Denoising $BSD68_{\sigma=25}$ | | Deblurring GoPro | | Low-Light LOLv1 | | Average | |
|---|---|---|---|---|---|---|---|---|---|---|---|---|---|---|
| NAFNet* (Chen et al., 2022a) | ECCV'22 | 17M | 25.23 | .939 | 35.56 | .967 | 31.02 | .883 | 26.53 | .808 | 20.49 | .809 | 27.76 | .881 |
| DGUNet* (Mou et al., 2022) | CVPR'22 | 17M | 24.78 | .940 | 36.62 | .971 | 31.10 | .883 | 27.25 | .837 | 21.87 | .823 | 28.32 | .891 |
| SwinIR* (Liang et al., 2021) | ICCVW'21 | 1M | 21.50 | .891 | 30.78 | .923 | 30.59 | .868 | 24.52 | .773 | 17.81 | .723 | 25.04 | .835 |
| Restormer* (Zamir et al., 2022) | CVPR'22 | 26M | 24.09 | .927 | 34.81 | .962 | 31.49 | .884 | 27.22 | .829 | 20.41 | .806 | 27.60 | .881 |
| MambaIR* (Guo et al., 2024a) | ECCV'24 | 27M | 25.81 | .944 | 36.55 | .971 | 31.41 | .884 | 28.61 | .875 | 22.49 | .832 | 28.97 | .901 |
| DL (Fan et al., 2019) | TPAMI'19 | 2M | 20.54 | .826 | 21.96 | .762 | 23.09 | .745 | 19.86 | .672 | 19.83 | .712 | 21.05 | .743 |
| Transweather | CVPR'22 | 38M | 21.32 | .885 | 29.43 | .905 | 29.00 | .841 | 25.12 | .757 | 21.21 | .792 | 25.22 | .836 |
| TAPE (Liu et al., 2022) | ECCV'22 | 1M | 22.16 | .861 | 29.67 | .904 | 30.18 | .855 | 24.47 | .763 | 18.97 | .621 | 25.09 | .801 |
| AirNet (Li et al., 2022) | CVPR'22 | 9M | 21.04 | .884 | 32.98 | .951 | 30.91 | .882 | 24.35 | .781 | 18.18 | .735 | 25.49 | .847 |
| IDR (Zhang et al., 2023) | CVPR'23 | 15M | 25.24 | .943 | 35.63 | .965 | 31.60 | .887 | 27.87 | .846 | 21.34 | .826 | 28.34 | .893 |
| PromptIR (Potlapalli et al., 2024) | NeurIPS'23 | 36M | 30.41 | .972 | 36.17 | .970 | 31.20 | .885 | 27.93 | .851 | 22.89 | .829 | 29.72 | .901 |
| MoCE-IR-S (Zamfir et al., 2025) | CVPR'25 | 11M | **31.33** | .978 | 37.21 | .978 | 31.25 | .884 | 27.80 | .877 | 21.68 | .851 | 30.08 | .913 |
| AdaIR (Cui et al., 2025) | ICLR'25 | 29M | 30.53 | .978 | 38.02 | .981 | 31.35 | .889 | 28.12 | .858 | 23.00 | .845 | 30.20 | .910 |
| MoCE-IR (Zamfir et al., 2025) | CVPR'25 | 25M | 30.48 | .974 | 38.04 | .982 | 31.34 | .887 | **30.05** | **.899** | 23.00 | .852 | 30.58 | **.919** |
| ExDA (Ours) | ICLR'26 | 22M | 31.14 | **.981** | **39.23** | **.985** | **31.52** | **.893** | 29.06 | .878 | **23.19** | **.859** | **30.83** | **.919** |
| Methods with the assistance of natural language prompts or multi-task learning | | | | | | | | | | | | | | |
| InstructIR-5D (Conde et al., 2024) | ECCV'24 | 16M | 36.84 | .973 | 27.10 | .956 | 31.40 | .887 | 29.40 | .886 | 23.00 | .836 | 29.55 | .908 |
| Art$_{PromptIR}$ (Wu et al., 2024) | ACM MM'24 | 36M | 29.93 | .908 | 22.09 | .891 | 29.43 | .843 | 25.06 | .765 | 21.99 | .811 | 25.81 | .846 |
| VLU-Net (Zeng et al., 2025) | CVPR'25 | 35M | 30.84 | .980 | 38.54 | .982 | 31.43 | .891 | 27.46 | .840 | 22.29 | .833 | 30.11 | .905 |
| Perceive-IR (Zhang et al., 2025) | TIP'25 | 42M | 28.19 | .964 | 37.25 | .977 | 31.44 | .887 | 29.46 | .886 | 22.88 | .833 | 29.84 | .909 |
| DFPIR (Tian et al., 2025) | CVPR'25 | 31M | 31.64 | .979 | 37.62 | .978 | 31.29 | .889 | 28.82 | .873 | 23.82 | .843 | 30.64 | .913 |
| DA-RCOT (Tang et al., 2025) | TPAMI'25 | 50M | 30.96 | .975 | 37.87 | .980 | 31.23 | .888 | 28.68 | .872 | 23.25 | .836 | 30.40 | .911 |

Table 3: *Comparison on **composited degradations***. PSNR (dB, ↑) and SSIM (↑) are reported.

| Method | Params. | CDD11-Single Low (L) | | Haze (H) | | Rain (R) | | Snow (S) | | CDD11-Double L+H | | L+R | | L+S | | H+R | | H+S | | CDD11-Triple L+H+R | | L+H+S | | Avg. | |
|---|---|---|---|---|---|---|---|---|---|---|---|---|---|---|---|---|---|---|---|---|---|---|---|---|
| AirNet | 9M | 24.83 | .778 | 24.21 | .951 | 26.55 | .891 | 26.79 | .919 | 23.23 | .779 | 22.82 | .710 | 23.29 | .723 | 22.21 | .868 | 23.29 | .901 | 21.80 | .708 | 22.24 | .725 | 23.75 | .814 |
| PromptIR | 36M | 26.32 | .805 | 26.10 | .969 | 31.56 | .946 | 31.53 | .960 | 24.49 | .789 | 25.05 | .771 | 24.51 | .761 | 24.54 | .924 | 23.70 | .925 | 23.74 | .752 | 23.33 | .747 | 25.90 | .850 |
| WGWSNet | 26M | 24.39 | .774 | 27.90 | .982 | 33.15 | .964 | 34.43 | .973 | 24.27 | .800 | 25.06 | .772 | 24.60 | .765 | 27.23 | .955 | 27.65 | .960 | 23.90 | .772 | 23.97 | .771 | 26.96 | .863 |
| WeatherDiff | 83M | 23.58 | .763 | 21.99 | .904 | 24.85 | .885 | 24.80 | .888 | 21.83 | .756 | 22.69 | .730 | 22.12 | .707 | 21.25 | .868 | 21.99 | .868 | 21.23 | .716 | 21.04 | .698 | 22.49 | .799 |
| OneRestore | 6M | 26.48 | .826 | 32.52 | .990 | 33.40 | .964 | 34.31 | .973 | 25.79 | .822 | 25.58 | .799 | 25.19 | .789 | 29.99 | .957 | 30.21 | .964 | 24.78 | .788 | 24.90 | .791 | 28.47 | .878 |
| MoCE-IR | 11M | 27.26 | .824 | 32.66 | .990 | 34.31 | .970 | 35.91 | .980 | 26.24 | .817 | 26.25 | .800 | 26.04 | .793 | 29.93 | .964 | 30.19 | .970 | 25.41 | .789 | 25.39 | .790 | 29.05 | .881 |
| ExDA (Ours) | 22M | **27.60** | **.837** | **34.44** | **.991** | **35.33** | **.973** | **36.85** | **.981** | **26.95** | **.834** | **26.80** | **.817** | **26.73** | **.809** | **31.17** | **.968** | **31.40** | **.972** | **25.59** | **.801** | **25.86** | .799 | **29.97** | **.891** |

Table 4: Comparisons for *4-task **adverse weather removal***. Missing values are denoted by '–'.

| Method | Venue | Snow100K-S PSNR | SSIM | Snow100K-L PSNR | SSIM | Outdoor-Rain PSNR | SSIM | RainDrop PSNR | SSIM | Average PSNR | SSIM |
|---|---|---|---|---|---|---|---|---|---|---|---|
| All-in-One (Li et al., 2020) | CVPR'20 | – | – | 28.33 | .882 | 24.71 | .898 | 31.12 | .927 | 28.05 | .902 |
| TransWeather (Valanarasu et al., 2022a) | CVPR'22 | 32.51 | .934 | 29.31 | .888 | 28.83 | .900 | 30.17 | .916 | 30.20 | .909 |
| Chen *et al.* (Chen et al., 2022b) | CVPR'22 | 34.42 | .947 | 30.22 | .907 | 29.27 | .915 | 31.81 | .931 | 31.43 | .925 |
| WGWSNet (Zhu et al., 2023) | CVPR'23 | 34.31 | .946 | 30.16 | .901 | 29.32 | .921 | 32.38 | .938 | 31.54 | .926 |
| WeatherDiff$_{64}$ (Özdenizci & Legenstein, 2023) | TPAMI'23 | 35.83 | .957 | 30.09 | .904 | 29.64 | .931 | 30.71 | .931 | 31.57 | .931 |
| WeatherDiff$_{128}$ (Özdenizci & Legenstein, 2023) | TPAMI'23 | 35.02 | .952 | 29.58 | .894 | 29.72 | .922 | 29.66 | .923 | 31.00 | .923 |
| AWRCP (Ye et al., 2023) | ICCV'23 | 36.92 | .965 | 31.92 | .934 | 31.39 | .933 | 31.93 | .931 | 33.04 | .941 |
| GridFormer (Wang et al., 2024) | IJCV'24 | 37.46 | .964 | 31.71 | .923 | 31.87 | .933 | 32.39 | .936 | 33.36 | .939 |
| MPerceiver (Ai et al., 2024) | CVPR'24 | 36.23 | .957 | 31.02 | .916 | 31.25 | .925 | **33.21** | .929 | 32.93 | .932 |
| DTPM (Ye et al., 2024) | CVPR'24 | 37.01 | .966 | 30.92 | .917 | 30.99 | .934 | 32.72 | .944 | 32.91 | .940 |
| Histoformer (Sun et al., 2024) | ECCV'24 | 37.41 | .966 | 32.16 | .926 | 32.08 | .939 | 33.06 | .944 | 33.68 | .944 |
| ExDA (Ours) | ICLR'26 | **37.97** | **.968** | **32.49** | **.930** | **32.70** | **.945** | 32.66 | **.944** | **33.92** | **.947** |

Tab. 2, our method achieves the highest average PSNR, surpassing even substantially larger models, as well as those leveraging additional modalities or pretraining. Moreover, for dehazing and deblurring, our approach also delivers competitive results compared to existing methods.

**Composited Degradation.** To better capture real-world complexities, we extend OneRestore (Guo et al., 2024b) to cover eleven scenarios, including rain, haze, snow, low-light conditions, and their composite variants. As summarized in Tab. 3, our approach consistently surpasses state-of-the-art methods such as AirNet (Li et al., 2022), PromptIR (Potlapalli et al., 2024), WGWSNet (Zhu et al., 2023), WeatherDiff (Özdenizci & Legenstein, 2023), OneRestore (Guo et al., 2024b), and MoCE-IR (Zamfir et al., 2025). Notably, our method achieves a 0.92 dB PSNR improvement over MoCE-IR, underscoring its strength in addressing complex, mixed degradations.

Table 5: Comparisons for Real-World *WeatherBench* (Guan et al., 2025).

| Method | Venue | Dehaze | | Derain | | Desnow | | Average | |
|---|---|---|---|---|---|---|---|---|---|
| | | PSNR | SSIM | PSNR | SSIM | PSNR | SSIM | PSNR | SSIM |
| Restormer (Zamir et al., 2022) | CVPR'22 | 19.30 | .687 | 34.48 | .945 | 27.95 | .836 | 27.25 | .823 |
| AirNet (Li et al., 2022) | CVPR'22 | 20.94 | .705 | 33.59 | .942 | 22.05 | .780 | 25.53 | .809 |
| TransWeather (Valanarasu et al., 2022a) | CVPR'23 | 19.79 | .680 | 29.34 | .903 | 24.96 | .796 | 24.70 | .793 |
| PromptIR (Potlapalli et al., 2024) | NeurIPS'23 | 21.11 | .713 | 34.53 | .944 | 27.93 | .836 | 27.86 | .831 |
| WGWS-Net (Zhu et al., 2023) | CVPR'23 | 13.79 | .603 | **37.08** | **.961** | 20.81 | .909 | 23.89 | .781 |
| Histoformer (Sun et al., 2024) | ECCV'24 | 17.69 | .669 | 30.70 | .916 | 25.39 | .808 | 24.59 | .798 |
| AdaIR (Cui et al., 2025) | ICLR'24 | 23.08 | .731 | 34.87 | .946 | 28.44 | .837 | 28.80 | .838 |
| ExDA (Ours) | ICLR'26 | **23.74** | **.739** | 35.86 | .946 | **29.42** | **.868** | **29.68** | **.851** |

Table 6: Comparisons for *3-task Medical Image Restoration*. Missing values are denoted by '−'.

| Method | Venue | MRI-Super-Resolution | | CT-Denoising | | PET-Synthesis | | Average | |
|---|---|---|---|---|---|---|---|---|---|
| | | PSNR | SSIM | PSNR | SSIM | PSNR | SSIM | PSNR | SSIM |
| Restormer (Zamir et al., 2022) | CVPR'22 | 31.72 | .936 | 33.61 | .918 | 37.13 | .947 | 34.16 | .934 |
| AirNet (Li et al., 2022) | CVPR'22 | 31.39 | .931 | 33.62 | .917 | 37.17 | .945 | 34.06 | .931 |
| Spach Transformer (Jang et al., 2023) | TMI'23 | 31.18 | .929 | 33.47 | .916 | 37.05 | .945 | 33.90 | .930 |
| DRMC (Yang et al., 2023) | MICCAI'23 | 29.55 | .903 | 33.28 | .915 | 36.19 | .945 | 33.00 | .930 |
| Eformer (Luthra et al.) | MedIA'23 | 29.19 | .873 | 32.44 | .908 | 35.11 | .909 | 32.25 | .897 |
| AMIR (Yang et al., 2024) | MICCAI'24 | 31.86 | .938 | **33.68** | .918 | 37.22 | .946 | 34.28 | .934 |
| ExDA (Ours) | ICLR'26 | **31.95** | **.938** | 33.67 | **.918** | **37.27** | **.947** | **34.30** | **.934** |

**Adverse Weather Removal.** Following prior works (Valanarasu et al., 2022b; Zhu et al., 2023), we evaluate ExDA on three challenging deweathering tasks: snow removal, rain streak and fog removal, and raindrop removal. Tab. 4 shows that ExDA consistently outperforms existing SOTA approaches across nearly all datasets, with the only exception being the PSNR result on RainDrop. These substantial improvements across diverse degradations highlight the robustness of ExDA . Especially, our method achieves a notable 0.24 dB PSNR gain over Histoformer (Sun et al., 2024).

**Real-world WeatherBench.** We further validate our approach on the challenging real-world WeatherBench benchmark (Guan et al., 2025), which involves diverse weather degradations such as haze, rain, and snow. As reported in Tab. 5, our method achieves the best overall performance, reaching an average PSNR/SSIM of 29.68/0.851. Notably, it delivers the highest scores on both dehazing and desnowing, and achieves competitive results in deraining compared with the strongest prior models. These consistent improvements across multiple degradations highlight the robustness and generalization ability of our method in handling complex, real-world weather conditions.

**Medical All-in-One.** Finally, we evaluate our method on medical image restoration tasks using the AMIR dataset (Yang et al., 2024), which unifies three restoration tasks within a single model. The results in Tab. 6 demonstrate the strong cross-domain applicability of our method, showing large gains in MRI super-resolution and competitive performance on the others.

**Visual Results.** Fig. 1 & 4 shows that our method restores high-quality and faithful results across diverse all-in-one settings, producing clearer boundaries and more accurate details than others.

## 4.2 ABLATION STUDY.

**Component Analysis & Model Scaling Exploration.** We start with two PromptIR-based baselines (with and without prompts). Unlike these settings, our framework does not depend on extra prompts for strong performance. Instead, the key improvements come from the proposed nonlinear value design and GST module. As shown in Fig. 5, adding a nonlinear value ("b + Nonlinear Value") provides a clear gain, and further integrating GST ("b + GST") yields consistent improvements. When combined ("c + d"), our full model achieves the best performance (32.96/0.921), confirming the effectiveness of these components. We then examine model scaling. Both the Small and Tiny versions ("e" and "f") remain highly competitive despite their reduced size, with the Tiny variant (only 6M parameters) still approaching the full model's performance. This demonstrates that once the core components are well designed, even extremely lightweight models can deliver strong results, making our framework both effective and scalable. The below ablations are all based on tiny model.

**Nonlinear Value Design.** To systematically investigate how nonlinear value transformations should be incorporated, we conduct ablations along three dimensions. *First*, we compare two formulations: an in-place nonlinear mapping ($V' = g_\theta(V)$) versus a residual form ($V' = V + g_\theta(V)$). Results consistently favor the residual design, which preserves the original representation while enriching it with nonlinear transformations.

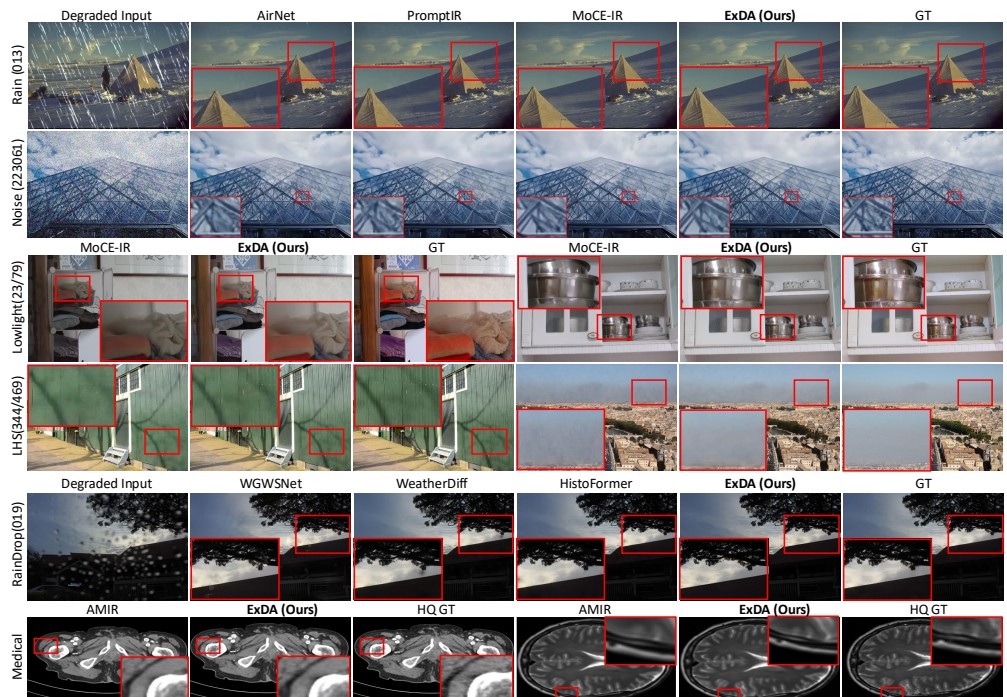

Figure 4: Visual results under diverse degradations and domains, spanning rain and noise (*3-deg*), low-light (*5-deg*), LHS composite (low-light+haze+snow), raindrop (adverse weather), and medical imaging. Please zoom in for details.

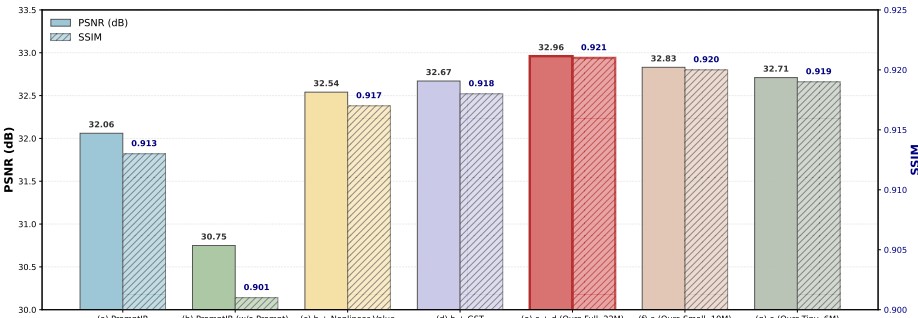

Figure 5: Qualitative results on medical restoration.

*Second*, we examine the choice of $g_\theta()$. While non-parametric nonlinearities (*e.g.*, pure Sigmoid or GELU) provide limited gains, learnable parameterized mappings achieve substantially better performance, confirming the need for adaptive flexibility. *Third*, we analyze where nonlinear values matter most in all-in-one blind IR by applying them selectively to the encoder, the decoder, or both. The evidence shows that deploying them in both the encoder and decoder yields the strongest improvements. Overall, these findings establish

Table 7: *Ablations* on **Nonlinear Value Design**

| Design Choice | Results | |
|---|---|---|
| | PSNR (dB, ↑) | SSIM (↑) |
| In-place Nonlinear (Sigmoid) | 32.30 | .913 |
| Residual Nonlinear (Sigmoid) | 32.63 | .916 |
| In-place Nonlinear (GELU) | 32.45 | .914 |
| Residual Nonlinear (GELU) | 32.71 | .919 |
| Non-parametric (GELU) | 32.30 | .914 |
| Parametric (Learnable) | 32.71 | .919 |
| Encoder-only | 32.65 | .916 |
| Decoder-only | 32.60 | .915 |
| Encoder + Decoder | 32.71 | .919 |

that residual, learnable, and fully integrated nonlinear value designs form the most effective strategy, significantly enhancing expressivity for diverse degradations in all-in-one IR (Tab. 7).

**GST Configuration.** We analyze the effect of the stride $s$ in GST, which controls the granularity of spatial information compression. When $s = 1$, the model preserves excessive redundancy and achieves 32.45 dB PSNR. Larger strides, such as $s = 4$ and $s = 8$, oversimplify spatial structures and reduce performance to 32.52 and 32.40 dB, respectively. The best result is obtained with $s = 2$, reaching 32.71 dB PSNR, striking the optimal balance between information preservation and compactness. Therefore, we adopt $s = 2$ as the default configuration in our method.

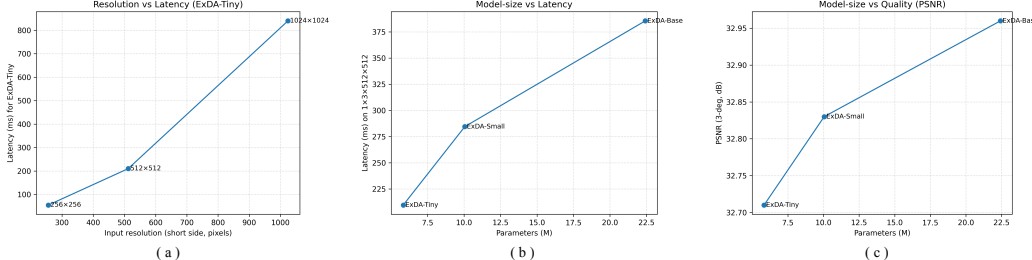

Figure 6: **Efficiency trade-offs of ExDA.** (a) Resolution–latency scaling (ExDA-Tiny). (b) Model-size–latency. (c) Model-size–quality (PSNR). ExDA-S provides the best accuracy–efficiency balance.

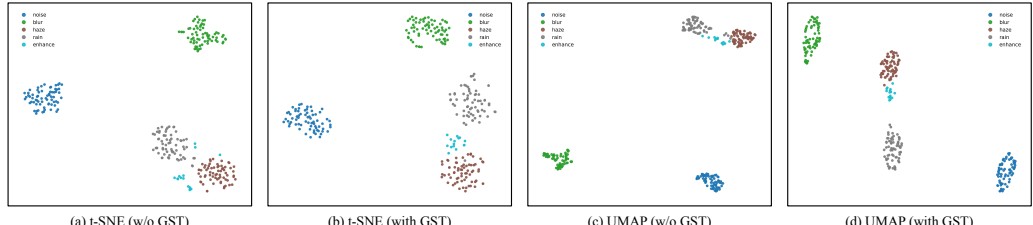

Figure 7: t-SNE and UMAP visualizations before (a,c) and after (b,d) applying GST.

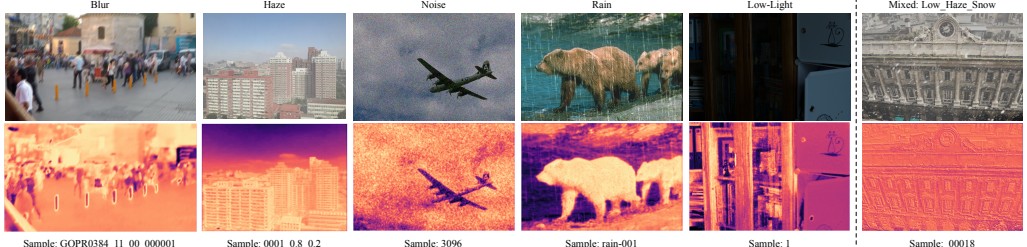

Figure 8: GST-driven attention visualization across 5 degradations and 1 mixed case. GST consistently highlights degradation-relevant regions, indicating that it captures meaningful degradation cues.

**Efficiency Analysis.** Fig. 6 summarizes the engineering trade-offs of ExDA-Tiny/Small/Base. Latency grows smoothly with model size (209.6→385.5 ms), while restoration quality increases correspondingly (32.71→32.96 dB). ExDA-Small provides the best balance. To illustrate scalability with input resolution, we sweep ExDA-Tiny from $256^2$ to $1024^2$, observing near-linear $O(HW)$ growth (54.5→840.1 ms). This confirms that ExDA scales predictably to high-resolution inputs.

**Degradation-Aware Representation.** As shown in Fig. 7, GST noticeably improves the organization of the embedding space: without GST, degradation types tend to overlap, while GST produces clearer and more compact clusters. To ensure a fair comparison given the uneven dataset sizes, we sampled 68 images for noise, blur, haze, and rain, and used all 15 samples for low-light. This balanced setting leads to substantial improvements in NMI (0.71→0.88) and ARI (0.56→0.89), reflecting stronger degradation-awareness. Fig. 8 further illustrates this effect by visualizing GST attention across blur, haze, noise, rain, low-light, and a mixed case (Low + Haze + Snow). Across all settings, GST consistently attends to degradation-relevant regions—such as blur boundaries, dense haze areas, noise-heavy textures, rain streaks, dark low-light structures, and mixed artifacts—showing that the module captures interpretable global cues that align with the improved embedding separation.

## 5 CONCLUSION

In this work, we revisited attention mechanisms for all-in-one image restoration and uncovered two central limitations in Restormer-style architectures: the linear constraint on value aggregation and the lack of explicit global context modeling. To address these issues, we introduced a nonlinear value transform and a global spatial token, two minimal yet powerful primitives that are both backbone-agnostic and lightweight. Our design consistently improves restoration quality across synthetic, real-world, underwater, and medical benchmarks, while remaining effective even in compact models with only a few million parameters. Beyond empirical gains, our analyses clarify why nonlinear value transformations enhance expressivity and why explicit global slots enable degradation-awareness. These insights highlight a broader principle: rethinking the role of values and global context is key to building robust, efficient, and general-purpose restoration models.

ACKNOWLEDGMENTS

This work was partially supported by the National Natural Science Foundation of China (No. 62473007), the Guangdong Provincial Key Laboratory of Ultra High Definition Immersive Media Technology (No. 2024B1212010006), the Shenzhen Innovation in Science and Technology Foundation for the Excellent Youth Scholars (No. RCYX20231211090248064), and the FIS project GUIDANCE (Debugging Computer Vision Models via Controlled Cross-modal Generation) (No. FIS2023-03251).

ETHICS STATEMENT

This work studies fundamental methods for all-in-one image restoration. Our research focuses on algorithmic improvements to attention mechanisms and does not involve the collection or annotation of new data. All datasets used in experiments (including synthetic, mixed, underwater, and medical benchmarks) are publicly available, widely used in the community, and employed strictly according to their licenses. The proposed methods are designed to improve robustness and generalization of image restoration models and do not raise foreseeable ethical concerns beyond the general considerations of responsible AI research.

REPRODUCIBILITY STATEMENT

We emphasize reproducibility throughout this work. Detailed descriptions of the architecture, training setups, and evaluation protocols are provided in the main text (see corresponding sections) and in the Appendix. If the paper is accepted, we will release all source code, pretrained weights, and configuration files immediately to facilitate transparent verification and future research.

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

## A  EXPERIMENTAL PROTOCOLS

### A.1  DATASETS

**3 Degradation Datasets.** For both the All-in-One and single-task settings, we follow the evaluation protocols established in prior works (Li et al., 2022; Potlapalli et al., 2024; Zamfir et al., 2025), utilizing the following datasets: For image denoising in the single-task setting, we combine the BSD400 (Arbelaez et al., 2010) and WED (Ma et al., 2016) datasets, and corrupt the images with Gaussian noise at levels $\sigma \in \{15, 25, 50\}$. BSD400 contains 400 training images, while WED includes 4,744 images. We evaluate the denoising performance on BSD68 (Martin et al., 2001) and Urban100 (Huang et al., 2015). For single-task deraining, we use Rain100L (Yang et al., 2020), which provides 200 clean/rainy image pairs for training and 100 pairs for testing. For single-task dehazing, we adopt the SOTS dataset (Li et al., 2018), consisting of 72,135 training images and 500 testing images. Under the All-in-One setting, we train a unified model on the combined set of the aforementioned training datasets for 130 epochs and directly test it across all three restoration tasks.

**5 Degradation Datasets.** The 5-degradation setting is built upon the 3-degradation setting, with two additional tasks included: deblurring and low-light enhancement. For deblurring, we adopt the GoPro dataset (Nah et al., 2017), which contains 2,103 training images and 1,111 testing images. For low-light enhancement, we use the LOL-v1 dataset (Wei et al., 2018), consisting of 485 training images and 15 testing images. Note that for the denoising task under the 5-degradation setting, we report results using Gaussian noise with $\sigma = 25$. The training takes 150 epochs.

**Composited Degradation Datasets.** Regarding the composite degradation setting, we use the CDD11 dataset (Guo et al., 2024b). CDD11 consists of 1,183 training images for: *(i) 4 kinds of*

*single-degradation types:* haze (H), low-light (L), rain (R), and snow (S); *(ii) 5 kinds of double-degradation types:* low-light + haze (l+h), low-light+rain (L+R), low-light + snow (L+S), haze + rain (H+R), and haze + snow (H+S). *(iii) 2 kinds of Triple-degradation type:* low-light + haze + rain (L+H+R), and low-light + haze + snow (L+H+S). We train our method for 150 epochs (fewer than 200 epochs than MoCE-IR (Zamfir et al., 2025)), and we keep all other settings unchanged.

**Adverse Weather Removal Datasets.** For the deweathering tasks, we follow the experimental setups used in TransWeather (Valanarasu et al., 2022a) and WGWSNet (Zhu et al., 2023), evaluating the performance of our approach on multiple synthetic datasets. We assess the capability of ExDA across three challenging tasks: snow removal, rain streak and fog removal, and raindrop removal. The training set, referred to as "AllWeather", is composed of images from the Snow100K (Liu et al., 2018), Raindrop (Qian et al., 2018), and Outdoor-Rain (Li et al., 2019) datasets. For testing, we evaluate our model on the following subsets: Snow100K-S (16,611 images), Snow100K-L (16,801 images), Outdoor-Rain (750 images), and Raindrop (249 images). Same as Histoformer (Sun et al., 2024), we train ExDA on "AllWeather" with 300,000 iterations.

**Real-World WeatherBench.** For the real-world WeatherBench, we adopt newly released benchmark from Guan et al. (2025), which contains in total 41,402 training pairs across 3 kinds of degradations (*i.e.*, rain, haze, and snow). There are 600 pairs for testing and 200 pairs per degradation. Under the All-in-One setting, we train a unified model on the combined set of the datasets for 120 epochs, and directly test it across all three restoration tasks

**Medical All-in-One Dataset.** AMIR dataset (Yang et al., 2024) include three important medical image restoration tasks, include *(i)* MRI super-resolution dataset from public LXI MRI benchmark. We use the public IXI MRI dataset, containing 578 high-quality T2-weighted MRI volumes. Low-quality images are generated by 4× k-space downsampling (retaining 6.25% of central data). The dataset is split 405/59/114 for training/validation/testing. *(ii)* CT Denoising dataset from NIH AAPM-Mayo Clinic Low-Dose CT Grand Challenge, These images originate from 10 patients, with 8 allocated for training, 1 for validation, and 1 for testing purposes, after slicing we get 18531/128/211 for training/validation/testing. *(iii)* PET Synthesis include 159 HQ PET images acquired by (Yang et al., 2024). The 3D volumes have dimensions of 192×192×400. Each volume is divided into 192 axial slices (192×400), excluding slices containing only air. Patient data is split into 120 training, 10 validation, and 29 testing cases.

## A.2 IMPLEMENTATION DETAILS

**Implementation Details.** Our ExDA framework is designed to be end-to-end trainable, removing the need for multi-stage optimization of individual components. The architecture adopts a robust 4-level encoder-decoder structure, with a varying number of Mixed Degradation Attention Blocks (MDAB) at each level—specifically $[3, 5, 5, 7]$ from highest to lowest resolution in the Tiny variant. Following prior works (Potlapalli et al., 2024; Zamfir et al., 2025), we train the model with a batch size of 32 in both the 3-Degradation All-in-One and single-task settings. The optimization uses a combination of $L_1$ and Fourier loss, optimized with Adam (Kingma & Ba, 2015) (initial learning rate of $2 \times 10^{-4}$, $\beta_1 = 0.9$, $\beta_2 = 0.999$) and a cosine decay schedule. During training, we apply random cropping to 128×128 patches, along with horizontal and vertical flipping as data augmentation. All experiments are conducted on a single NVIDIA H200 GPU (140 GB). Memory usage is approximately 42 GB for the Tiny (*i.e.*, ExDA -T) model and 56 GB for the Small model (*i.e.*, ExDA -S).

## A.3 OPTIMIZATION OBJECTIVES

The overall optimization objective of our approach is defined as:

$$\mathcal{L}_{\text{total}} = \mathcal{L}_1 + \lambda_{fre} \times \mathcal{L}_{\text{Fourier}} \tag{4}$$

Here, $\mathcal{L}_{\text{Fourier}}$ denotes the real-valued Fourier loss computed between the restored image and the ground-truth image, and $\mathcal{L}_{\text{SPD}}$ represents our proposed contrastive learning objective in the SPD (Symmetric Positive Definite) space.

Specifically, we adopt an $\ell_1$ loss that adopted in IR tasks (Potlapalli et al., 2024; Zamfir et al., 2025; Li et al., 2022; Ren et al., 2026; Cui et al., 2025; Ren et al., 2024; Li et al., 2025), defined as $\mathcal{L}_1 = |\hat{x} - x|_1$, to enforce pixel-wise similarity between the restored image $\hat{x}$ and the ground-truth image $x$. $\mathcal{L}_{\text{Fourier}}$, as utilized in MoCE-IR (Zamfir et al., 2025; Cui et al., 2025), to enhance

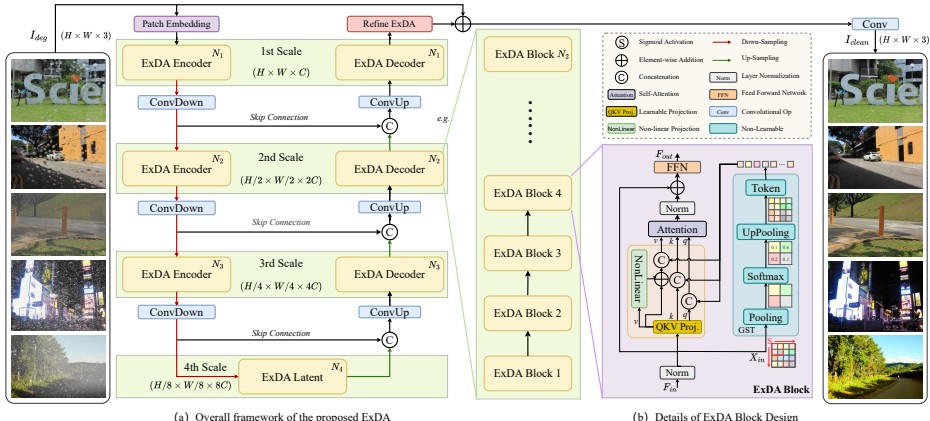

(a) Overall framework of the proposed ExDA  (b) Details of ExDA Block Design

Figure A: ExDA Model Architecture Framework. (a) Demonstrates the overall pipeline for ExDA, (b) details the ExDA block design.

Table A: The details our the tiny and small version of our ExDA . FLOPs are computed on an image of size $224 \times 224$ using a NVIDIA Tesla A100 (40G) GPU.

|  | ExDA -Base | ExDA -S | ExDA -T |
|---|---|---|---|
| The Number of the MDAB crosses 4 scales | [4, 6, 6, 8] | [4, 6, 6, 8] | [3, 5, 5, 7] |
| The Input Embedding Dimension | 48 | 32 | 26 |
| The FFN Expansion Factor | 2 | 2 | 2 |
| The Number of the Refinement Blocks | 4 | 3 | 3 |
| Params. ($\downarrow$) | 22.42M | 10.05M | 5.83 M |
| FLOPs ($\downarrow$) | 109G | 47 G | 27 G |

frequency-domain consistency, the real-valued Fourier loss, is defined as:

$$\mathcal{L}_{\text{Fourier}} = \|\mathcal{F}_{\text{real}}(\hat{x}) - \mathcal{F}_{\text{real}}(x)\|_1 + \|\mathcal{F}_{\text{imag}}(\hat{x}) - \mathcal{F}_{\text{imag}}(x)\|_1 \,, \tag{5}$$

where $\hat{x}$ and $x$ denote the restored and ground-truth images, respectively. $\mathcal{F}_{\text{real}}(\cdot)$ and $\mathcal{F}_{\text{imag}}(\cdot)$ represent the real and imaginary parts of the 2D real-input FFT (*i.e.*, rfft2). The final loss is computed as the $\ell_1$ distance between the real and imaginary components of the predicted and target frequency spectra. Same as MoCE-IR (Zamfir et al., 2025), $\lambda_{fre}$ is set to 0.1 throughout our experiments.

## B MACRO ARCHITECTURE INTRODUCTION

The overall architecture of ExDA is illustrated in Fig. A. At a macro level, it adopts a U-shaped network with four hierarchical levels. Initially, a convolutional layer extracts shallow features from the degraded input, creating a patch embedding of size $H \times W \times C$. As in standard U-Nets, each encoder stage doubles the embedding dimension and halves the spatial resolution, with skip connections transferring information to the corresponding decoder stage. In the decoder, features are merged with the previous decoding stage via linear projection. Finally, a global skip connection links input to output, preserving high-frequency details and producing the restored image.

Unlike most recent approaches that rely on additional prompts or large language models (LLMs), we argue that a simple encoder–decoder architecture, as illustrated in Fig. 1(I)(d), is sufficient to tackle the challenges of all-in-one image restoration. The key lies in identifying and addressing the latent bottlenecks.

**Model Scaling.** We propose two scaled variants of our ExDA , namely Tiny (ExDA -T) and Small (ExDA -S). As detailed in Tab. A, these variants differ in terms of the number of MDAB blocks across scales, the input embedding dimension, the FFN expansion factor, and the number of refinement blocks.

**Efficiency Comparison.** Tab. A presents a detailed comparison of PSNR, memory usage, parameter count, and FLOPs. Our Tiny model (ExDA -T) achieves the best efficiency-performance trade-off: with only 6.21M parameters and 16G FLOPs, it outperforms all prior methods, including larger

models like PromptIR (Potlapalli et al., 2024) and MoCE-IR-S (Zamfir et al., 2025). Notably, ExDA -T surpasses MoCE-IR-S by +0.26 dB while requiring less than half the computational cost. Even our Small variant (ExDA -S) exceeds full MoCE-IR in both PSNR (+0.18 dB) and FLOPs (27G vs. 75G). These results validate that our design achieves strong restoration quality without sacrificing computational efficiency.

## C  ADDITIONAL EXPLANATION OF THE NONLINEAR VALUE PATH

This section provides a more detailed explanation of the operator-level effect and motivation behind introducing a nonlinear value transform into Restormer-style channel-wise attention.

**Structural limitation of linear-value attention.** As discussed in Sec. 3.1 of the main paper, when the value projection is strictly linear, the attention operator is fundamentally constrained to lie within the linear span of the input features $\mathcal{X}$. For any attention head, we may write the attention operator as:

$$\mathrm{Attn}(Q, K, V) = AV, \qquad A = \mathrm{Softmax}(QK^\top). \tag{6}$$

Since $V$ is a linear transformation of $\mathcal{X}$, its output is restricted by:

$$\mathrm{Range}(\mathrm{Attn}) \subseteq \mathrm{span}(\mathcal{X}), \tag{7}$$

a limitation that cannot be removed by widening the FFN or adding post-aggregation nonlinearities, as these operations act *after* the linear mixing in Eq. 6.

**Effect of the residual nonlinear value transform.** To relax this structural constraint, the proposed module applies a residual non-affine transform to the value branch:

$$V' = V + g_\theta(V), \tag{8}$$

where $g_\theta$ includes depth-wise convolutions and a GELU activation. Because $g_\theta$ is non-affine, we have:

$$g_\theta(\mathcal{X}) \not\subseteq \mathrm{span}(\mathcal{X}), \tag{9}$$

which enlarges the admissible value subspace:

$$\mathrm{span}(\mathcal{X}) \subsetneq \mathrm{span}\big(\mathcal{X} \cup g_\theta(\mathcal{X})\big). \tag{10}$$

This is an operator-level property that directly increases the functional rank of the attention mechanism, independent of the global universal-approximation capacity of the entire network.

**Empirical validation.** Sec. 3.2 of the main paper provides evidence of the limitations of linear-value attention through synthetic function fitting and MNIST restoration experiments.

## D  ADDITIONAL ANALYSIS

To complement the quantitative stride ablation in the main paper, we further visualize in Fig. B how the GST-driven attention maps change with different stride values $s \in \{1, 2, 4, 8\}$ across the five degradations (noise, blur, haze, rain, low-light). A clear trend emerges: (i) $s\!=\!1$ preserves the most spatial detail but also introduces redundant high-frequency responses; (ii) $s\!=\!2$ maintains fine structures while producing clean and well-localized degradation cues; (iii) larger strides ($s\!=\!4$ and $s\!=\!8$) overly downsample the spatial statistics and yield noticeably blurred or weakened attention responses. These observations align with our quantitative findings, showing that $s\!=\!2$ provides the best balance between preserving degradation-relevant information and avoiding redundancy.

## E  ADDITIONAL VISUAL RESULTS

### E.1  3-DEGRADATION

We additionally put more visual results on 3 Degradation settings, please refer to Fig. C for more detailed information.

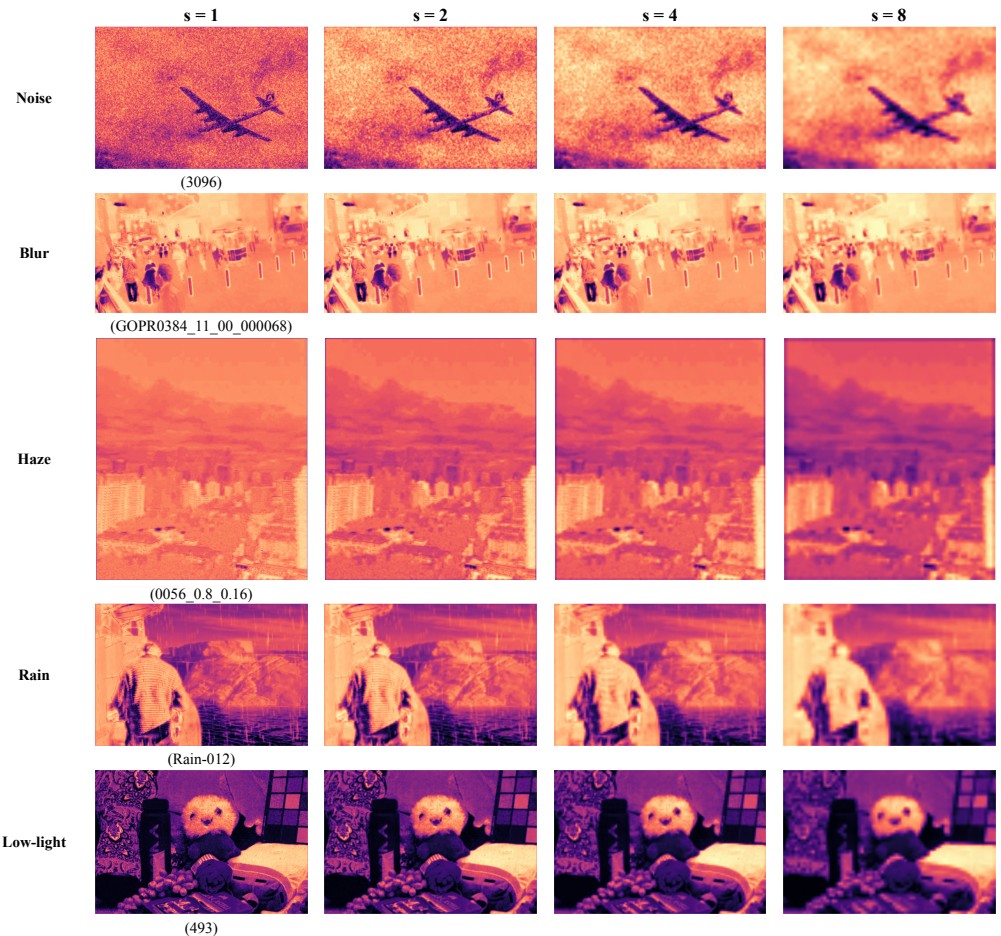

Figure B: Qualitative comparison of GST-driven attention maps across different stride values $s$. For five degradations (noise, blur, haze, rain, and low-light), we visualize the GST heatmaps obtained with $s \in \{1, 2, 4, 8\}$. Smaller strides preserve finer spatial cues, while larger strides yield overly smoothed responses. Please zoom in for more details.

## E.2   5-DEGRADATION

We additionally put more visual results on 5 Degradation settings, please refer to Fig. D for more detailed information.

## E.3   COMPOSITED DEGRADATION

We additionally put more visual results on Composited Degradation settings, please refer to Fig. E for more detailed information. Here, we selected the scenes composited by lowlight, haze and snow or lowlight, haze and rain, which are the most difficult settings. Our results indicated that our method can easily handle difficult and severe weather degradation.

## E.4   ADVERSE WEATHER REMOVAL

We additionally put more visual results on real-world WeatherBench data, please refer to Fig. F for more detailed information. Our results indicated that our method can handle different difficult and severe weather degradations.

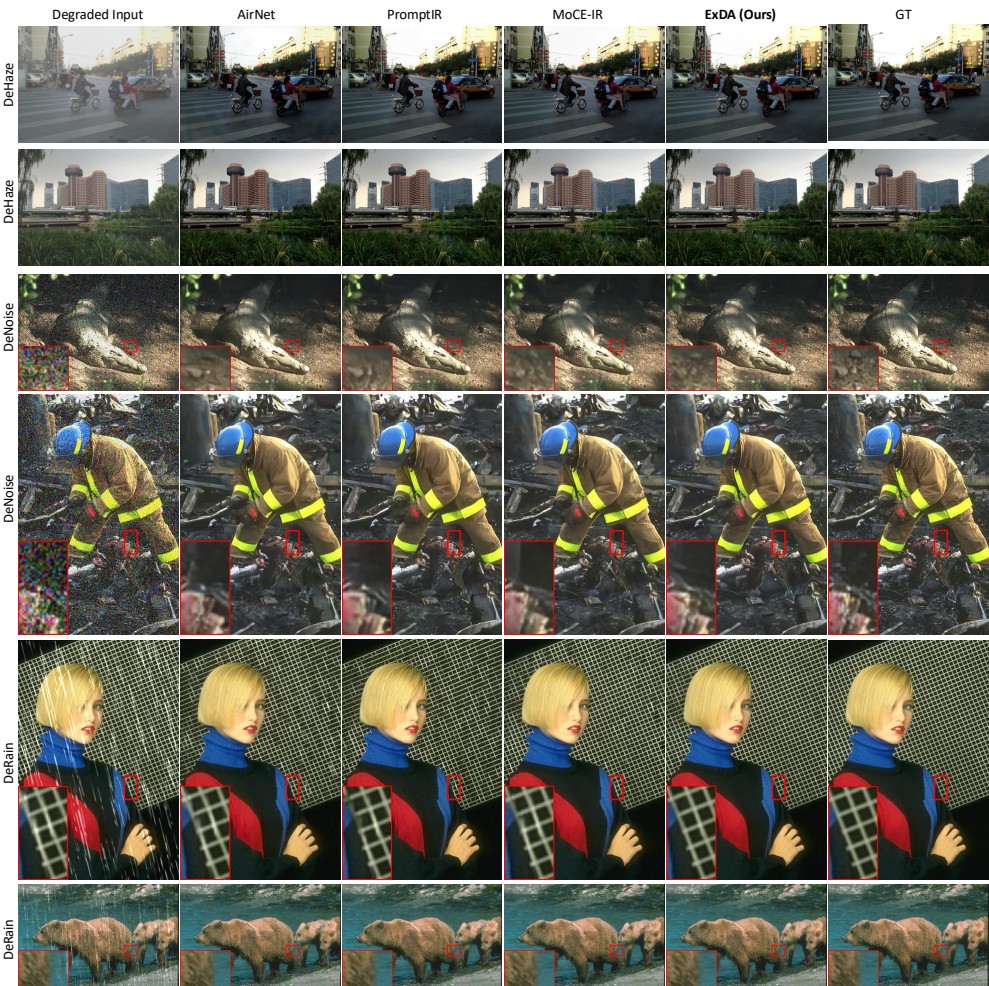

Figure C: More qualitative comparisons under 3-deg setting. Zoom in to see more details.

### E.5 REAL-WORLD WEATHERBENCH

We additionally put more visual results on real-world WeatherBench data, please refer to Fig. G for more detailed information. Our results indicated that our method can easily handle difficult and severe real-world weather degradation.

### E.6 MEDICAL IMAGE RESTORATION

We provide additional visual results for medical image restoration. For better visualization, we display 64x64 crops. Both our method and the state-of-the-art (SOTA) baseline, AMIR, effectively enhance image quality. However, our approach achieves better detail, producing clearer tissue structures and sharper boundaries.

## F   USE OF LARGE LANGUAGE MODELS (LLMS).

Use of Large Language Models (LLMs). We used a large language model (ChatGPT) solely to aid in polishing the writing and improving the readability of the manuscript, such as refining grammar, style, and clarity. The model was not involved in idea generation, experimental design, implementation, analysis, or any other research-related aspects of this work. All scientific contributions, including the problem formulation, methodology, experiments, and conclusions, are entirely the responsibility of the authors.

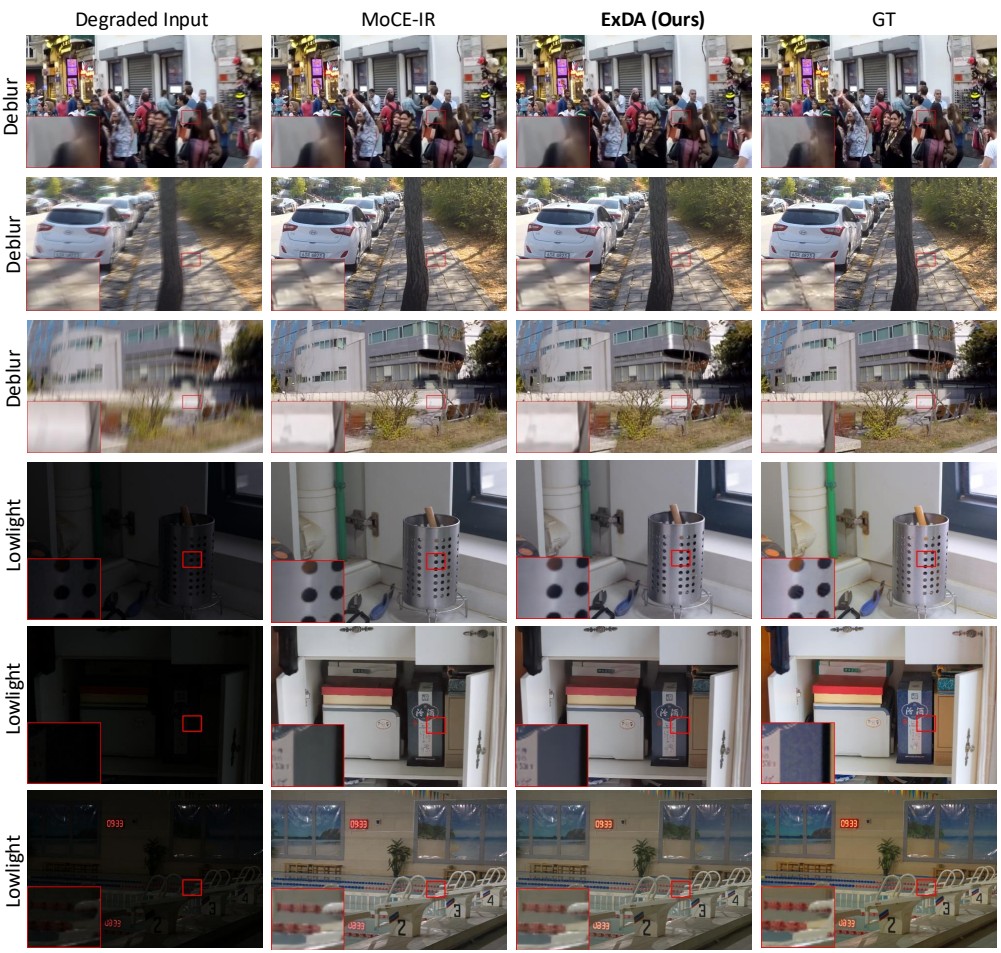

Figure D: More qualitative comparisons under 5-deg setting. Zoom in to see more details.

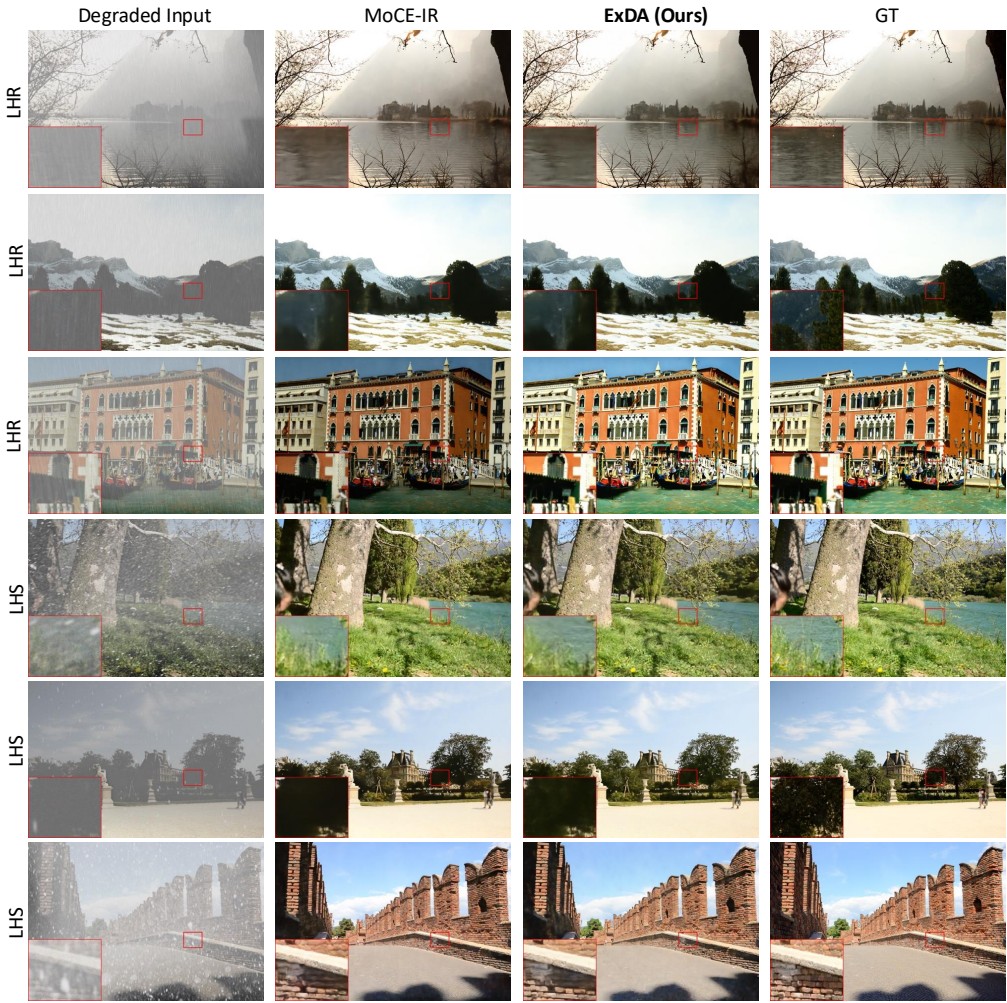

Figure E: More qualitative comparisons under composited degradation setting. Zoom in to see more details.

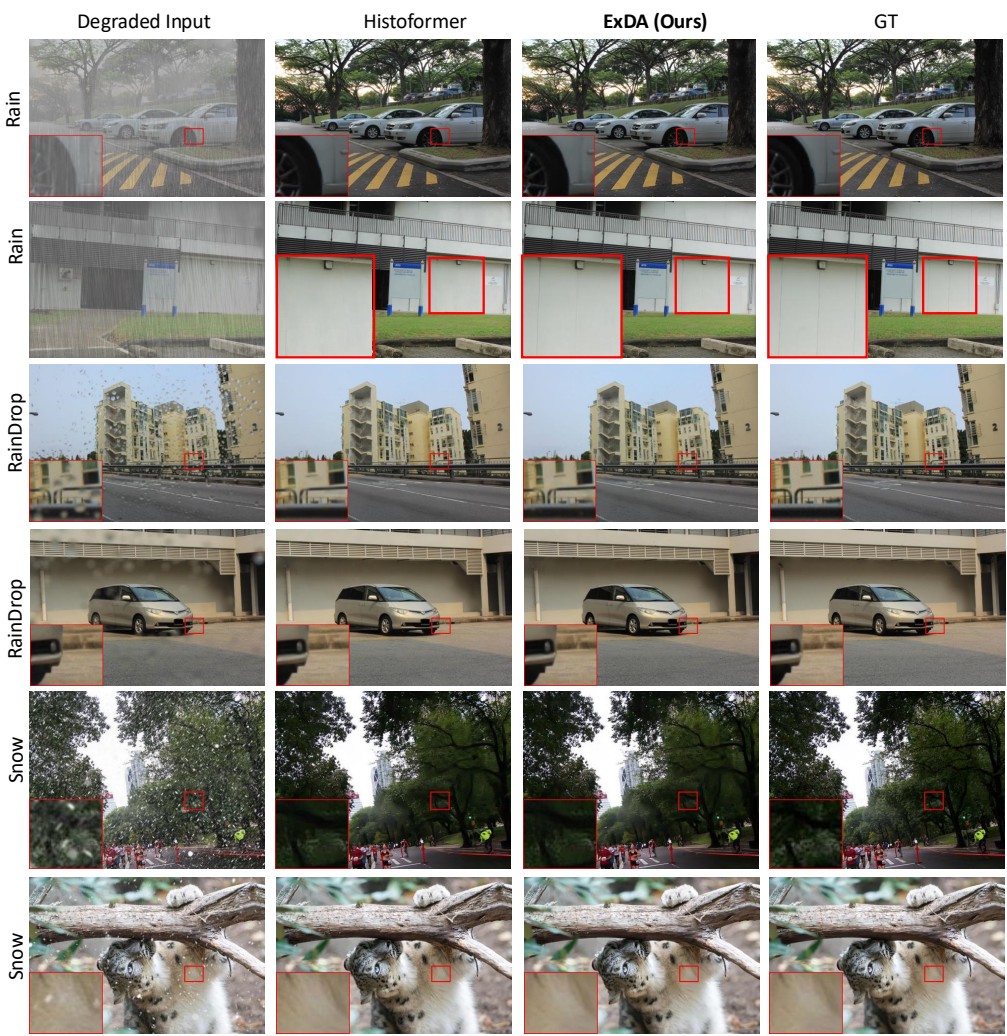

Figure F: More qualitative comparisons under adverse weather. Zoom in to see more details.

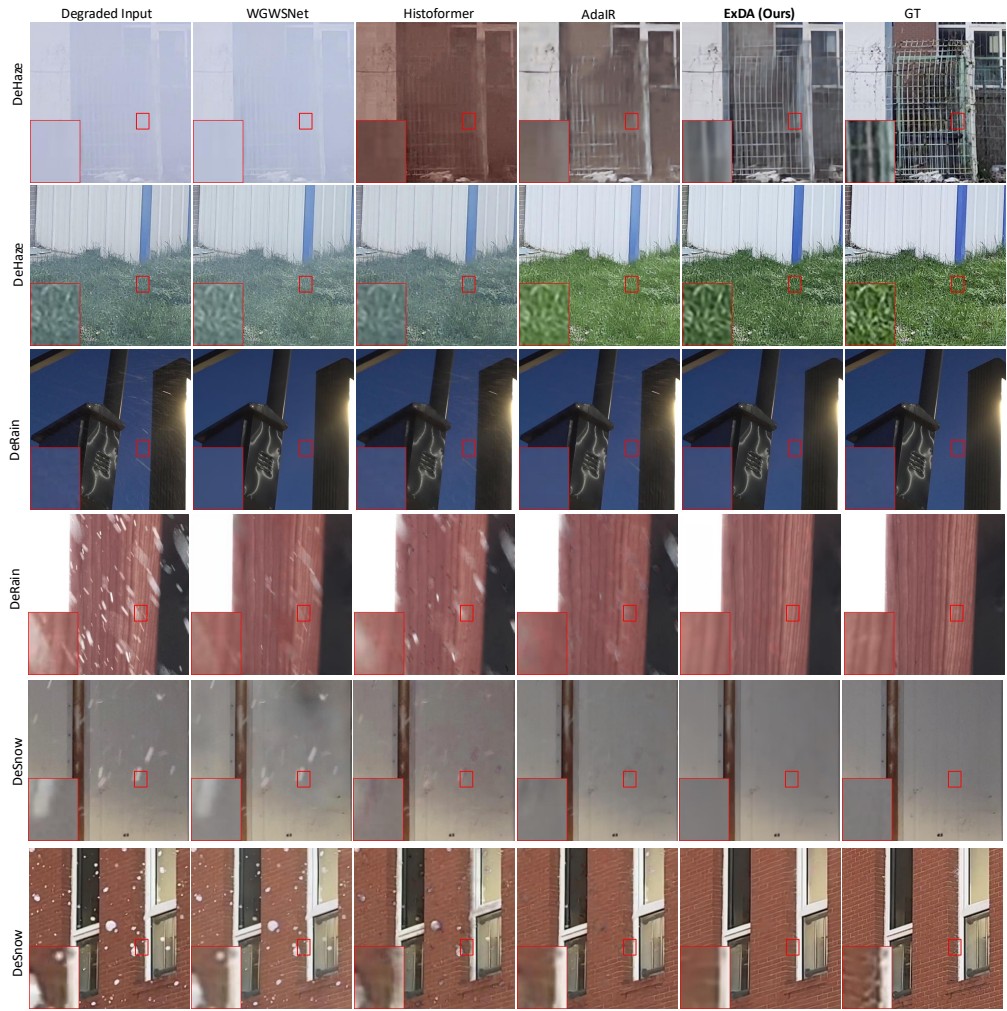

Figure G: More qualitative comparisons under real-world weatherbench data. Zoom in to see more details.

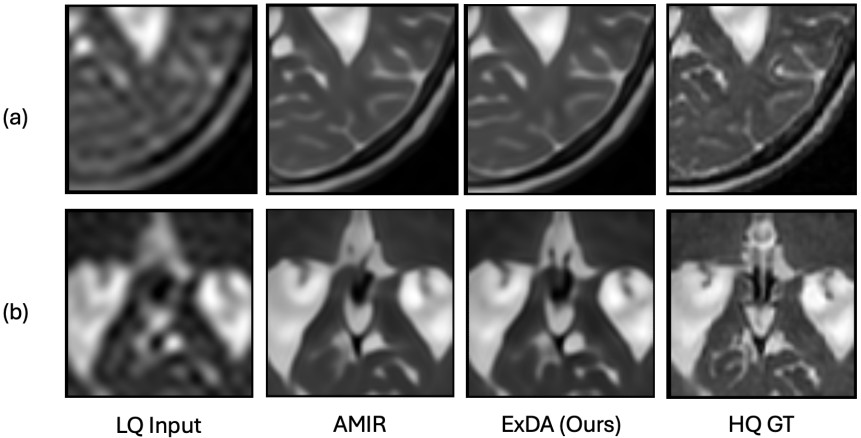

Figure H: More qualitative comparisons under medical restoration MRI super-resolution setting.

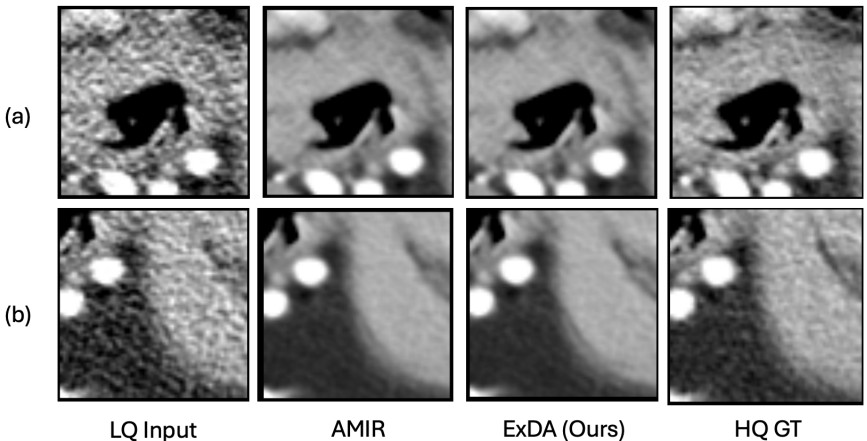

Figure I: More qualitative comparisons under medical restoration CT-Denoising setting.

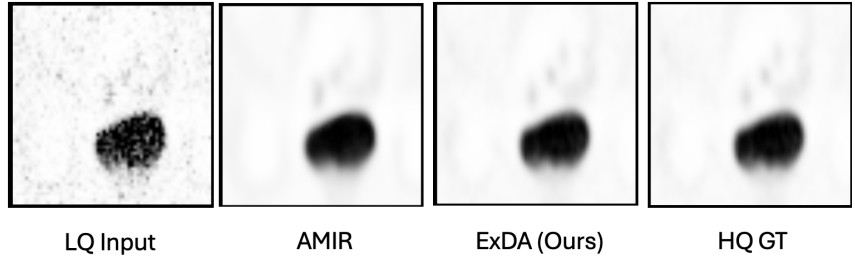

Figure J: More qualitative comparisons under medical restoration PET-Synthesis setting.

