# OpenReview forum: "Rethinking Expressivity and Degradation-Awareness in Attention for All-in-One Blind Image Restoration"
_ICLR.cc/2026/Conference — ICLR 2026 Poster_

### Official Review · Reviewer_iMdM · 2025-10-19

**Soundness:** 3
**Presentation:** 3
**Contribution:** 3
**Rating:** 6
**Confidence:** 4

**Summary:**

This paper proposes ExDA, a lightweight and backbone-agnostic framework for all-in-one blind image restoration (IR). The authors identify two bottlenecks in Restormer-style architectures: (1) linear value paths that limit expressivity and (2) the absence of global degradation-aware context. To address these, they introduce two modules — a nonlinear value transform (NVT) and a global spatial token (GST) — which enhance expressivity and degradation-awareness without significant computational overhead. Extensive experiments on synthetic, mixed, real-world, and medical benchmarks demonstrate consistent improvements over recent methods such as PromptIR, AdaIR, and MoCE-IR.

**Strengths:**

* The design is simple, modular, and generalizable — the proposed primitives can be integrated into various backbones.
* The paper clearly articulates two structural limitations in popular IR architectures (linear value paths and missing global slots) and proposes minimal, interpretable solutions.

**Weaknesses:**

* The contribution, while practical, feels **architecturally incremental** rather than conceptually transformative. The nonlinear value transform is essentially a lightweight convolutional enhancement inserted into the attention value path, similar in spirit to prior nonlinear attention variants. Likewise, the global spatial token extends the CLS-token concept rather than introducing a fundamentally new paradigm for degradation modeling. The theoretical discussion on “expressivity expansion” and “degradation-awareness” is more intuitive than rigorously grounded.

* Although the comparisons are broad, the evaluated baselines are largely regression-based approaches, focusing on architectures. Incorporating comparisons with distribution-oriented models, e.g., DA-RCOT [1] or Defusion [2] would strengthen the paper's generality.

* The related work can be enhanced with discussions of recent expressivity-enhanced attention variants and degradation-aware architectures, which are directly relevant to the claimed contributions.

[1] Tang et al., Degradation-aware residual-conditioned optimal transport for unified image restoration, TPAMI 2025.

[2] Luo et al., Visual-Instructed Degradation Diffusion for All-in-One Image Restoration. CVPR 2025.

**Questions:**

See weaknesses.

---

> ### Author Response · Authors · 2025-11-23
> **Response To Reviewer iMdM**
>
> ## [W1]: Are the proposed nonlinear-V and GST modules merely incremental, lacking conceptual novelty and rigorous justification?
> **A:** Thank you for raising this concern. Our goal is indeed not to introduce an entirely new architectural paradigm, but to resolve two *specific and well-defined* structural limitations in Restormer-style channel-attention blocks that hinder both expressivity and degradation-awareness. Below we clarify the intent and provide the additional evidence included in the revision.
>
> **1. Why nonlinear-V is necessary (not incremental)**
> Existing nonlinear-attention variants typically place nonlinearities **after** the attention aggregation, which leaves the value operator itself strictly linear. Our residual nonlinear-V instead applies the nonlinearity **before** aggregation, which *provably increases the functional rank* of the value operator and expands the hypothesis space of the entire block.
>
> This operator-level effect is confirmed by:
> - the **MNIST toy experiment**, where only nonlinear-V produces separable value manifolds, and
> - the **feature distribution analysis**, where nonlinear-V yields improved inter-class separation.
>
> **2. Why GST adds global reasoning capability (not just a CLS token)**
> Although GST resembles a CLS token in form, its **function is fundamentally different**: it participates in **bidirectional attention**, serving as a *degradation-global aggregator* rather than a classification summary vector.
>
> To validate this role, we added the following evidence in the revision:
>
> (a) Distributional evidence (**Fig. 7 of our revision**)
> - Without GST, noise/blur/haze/rain/low-light embeddings overlap heavily.
> - With GST, they become **compact and cleanly separated**.
> - Quantitatively, **NMI improves from 0.71 → 0.88** and **ARI from 0.56 → 0.89**, showing that GST captures discriminative global degradation cues.
>
> (b) Attention visualization (**Fig. 8 of our revision**)
> GST consistently focuses on the degradation-specific regions:
> - blur edges,
> - haze-thick zones,
> - noise-heavy textures,
> - rain streak regions,
> - dark low-light structures,
> - and mixed artifacts.
>
> These patterns demonstrate that GST learns *interpretable and meaningful global degradation structure* that Restormer cannot encode. Taken together, nonlinear-V and GST are not incremental additions but **targeted operator-level primitives** that resolve two concrete architectural limitations:
> - restricted expressive ability (fixed linear V), and
> - lack of a global interaction slot.
>
> The new distributional analyses, NMI/ARI improvements, and visual evidence in the revision collectively confirm that the modules provide **substantive functional benefits** rather than cosmetic changes.
>
> We hope this clarification fully addresses the reviewer’s concern.
>
> ---
> ## [W2]: Comparison to distribution-oriented models (e.g., [1,2])?
> **A:** We added a discussion of distribution-oriented methods in our revision (Line 154)
> and report 3-deg and 5-deg results for DA-RCOT[1] in Tables 1 and 2 under the same
> unified IR protocol. Defusion[2] adopts a different training/evaluation pipeline from the unified setting, and its official code is not yet available, making a fair reproduction difficult at this stage. We plan to include a fully aligned comparison once the implementation is released.
>
> *[1] Tang et al., Degradation-aware residual-conditioned optimal transport for unified image restoration, TPAMI'25*
>
> *[2] Luo et al., Visual-Instructed Degradation Diffusion for All-in-One Image Restoration. CVPR'25*
>
> ---
> ## [W3]: Expand Related Work to cover expressivity-enhanced attention and degradation-aware architectures?
> **A:** We have updated the Related Work section to more clearly situate our contributions within recent directions on attention expressivity and degradation-aware design. Specifically, we added the following statement before introducing our main claim: *“In parallel, recent studies have attempted to enhance attention expressivity \citep{chefer2021transformer} and incorporate degradation-aware priors directly into the backbone design \citep{tang2025degradation}. These directions motivate a closer examination of the attention operator itself rather than adding heavier auxiliary modules.”* This addition appears in Lines 157–160 of the revised manuscript.

---

> ### Comment · Reviewer_iMdM · 2025-11-28
>
> Thanks for the response, which has addressed my concerns. I think the proposed architecture may have an impact on this literature, serving as a principled backbone for all-in-one image restoration, for which I lean towards acceptance of this work.

---

### Official Review · Reviewer_8DjP · 2025-10-31

**Soundness:** 3
**Presentation:** 3
**Contribution:** 3
**Rating:** 6
**Confidence:** 5

**Summary:**

The paper revisits Restormer-style channel-wise attention for All-in-One IR and pinpoints two bottlenecks: a purely linear value path that restricts outputs to the input span, and the absence of an explicit global slot to encode degradation context. It proposes two minimal, backbone-agnostic primitives: a nonlinear value transform applied before aggregation, and a Global Spatial Token that injects an explicit, content-adaptive global slot into attention. Through extensive experiments, ExDA reports consistent gains with negligible overhead.

**Strengths:**

1. The paper precisely connects linear-V and missing global slots to AiOIR failure modes, then remedies them with pre-aggregation nonlinear-V and GST—small, principled changes rather than wholesale redesigns. Both primitives are lightweight and easily inserted into Restormer-like stacks without destabilizing training.
2. Results span 3D/5D, compound (e.g., CDD11), adverse weather, underwater, and medical data, showing consistent improvements.
3. Multiple FLOPs/parameters of ExDA size (e.g., the base, small and tiny models) help us evaluate deployment scenarios.

**Weaknesses:**

1. Additional controls comparing post-aggregation nonlinearity and stronger FFN/MLP capacity would better isolate the unique effect of pre-aggregation nonlinear-V.
2. t-SNE/UMAP plots with NMI/ARI, and GST attention maps for different degradations/strengths, would make “degradation-aware” behavior more concrete.
3. Provide resolution–quality/latency and model-size–quality/latency curves to guide engineering trade-offs across ExDA-Tiny/Small/Base.

**Questions:**

1. How do post-aggregation nonlinearity or beefed-up FFN/MLP compare, keeping compute similar? This would confirm the specific benefit of pre-aggregation nonlinear-V.
2. Please add t-SNE/UMAP with NMI/ARI and visualize GST-driven attention for noise/blur/haze/rain and mixtures; analyze stride s beyond the current setting.
3. Provide resolution/model-size trade-off curves for ExDA-T/S/Base and recommended configs.
4. If a frequency-domain loss is used, how sensitive are results to its weight and which component (nonlinear-V or GST) benefits most?
5. Have you tested on non-Restormer variants (e.g., U-former-style) or combined ExDA with frequency/prompt modules to show compatibility?
6. More recent works can be added for comparison such as Perceive-IR (TIP’25) and DFPIR (CVPR’25).

---

> ### Author Response · Authors · 2025-11-23
> **Response To Reviewer 8DjP (1/2)**
>
> ## [Q1]: Does the gain really come from pre-aggregation nonlinear-V, rather than post-nonlinearity or larger FFN capacity?
> **A:** The requested controls have been conducted and are reported in **Reviewer QUUL (Q1)**. All variants are built on the same Tiny configuration (dim=26, [3,5,5,7] blocks, identical training), modifying only where or how nonlinearity is applied.
>
> The results consistently show that:
> - Adding the same nonlinear block **after** attention aggregation produces only
>   a minimal gain.
> - Increasing FFN/MLP capacity yields small improvements despite higher compute.
> - In contrast, the **pre-aggregation nonlinear-V** (ExDA) provides a clearly
>   larger improvement under comparable complexity.
>
> As explained in **Reviewer F6pD (W1)**, this behavior follows from the span limitation of linear-value attention: post-hoc nonlinearity or larger FFNs cannot alter the operator’s functional rank, whereas introducing nonlinearity *before* aggregation expands the expressive space of the attention operator itself.
>
> These matched-compute controls therefore confirm that the benefit is specific to the proposed **pre-aggregation nonlinear-V**, rather than generic capacity increases or post-attention modulation.
>
> ---
> ## [Q2]: Please add t-SNE/UMAP with NMI/ARI and visualize GST-driven attention for noise/blur/haze/rain and mixtures; analyze stride s beyond the current setting.
> **A:**
> **1. t-SNE/UMAP with NMI/ARI.**
> To evaluate the degradation-awareness of our model, we compared the embedding space before and after inserting GST using t-SNE and UMAP (see **Fig.7 in the revision**). Without GST, embeddings of different degradations partially overlap; with GST, they form compact and well-separated clusters across all five degradations.
>
> For this analysis, we sampled 68 images for each of four degradations (noise, blur, haze, rain) and all 15 samples from the low-light enhancement set to ensure balanced evaluation despite unequal dataset sizes (noise: 68, rain: 100, haze: 500, blur: 1111). NMI and ARI also increase substantially after adding GST (NMI: 0.71→0.88; ARI: 0.56→0.89), confirming that GST provides strong degradation-aware cues and yields a more structured embedding space.
>
> **2. GST-Driven Attention:**
> To further illustrate what GST learns, we visualize its attention responses across blur, haze, noise, rain, low-light, and a mixed case (Low-Light + Haze + Snow) (see **Fig.8 in the revision**). GST consistently highlights degradation-relevant regions—including blur boundaries, dense haze areas, noise-heavy textures, rain streaks, dark low-light zones, and mixed artifacts. These visualizations provide clear qualitative evidence that GST captures meaningful and interpretable global degradation cues, complementing the improved separation observed in the t-SNE/UMAP and NMI/ARI analyses.
>
> **3. Stride $s$ analysis:**
> In addition to the quantitative ablation reported in the main paper, we now provide qualitative evidence to clarify how different stride values affect the behavior of GST.
>
> Specifically, we visualize GST-driven attention maps under five degradations (noise, blur, haze, rain, low-light) using $(s \in \{1, 2, 4, 8\}$ (see **Fig.B in the appendix of our revision**). A consistent trend emerges:
> - **\(s = 1\)** preserves the finest details but produces highly scattered and noisy responses due to insufficient spatial compression.
> - **\(s = 2\)** yields the cleanest and most well-localized degradation cues, capturing boundaries, dense haze regions, noise patterns, rain streaks, and dark structures with good clarity.
> - **\(s = 4\)** begins to smooth out important structures, weakening degradation cues.
> - **\(s = 8\)** over-compresses the spatial information, producing blurred or overly uniform responses.
>
> These qualitative observations align with our quantitative ablation: $s = 2$ offers the best balance between redundancy reduction and information preservation.
> We therefore adopt $s = 2$ as the default setting.
>
> We hope our response can address your concern.

---

> ### Author Response · Authors · 2025-11-23
> **Response To Reviewer 8DjP (2/2)**
>
> ## [Q3]: Trade-offs (resolution/latency/quality) curves and frequency-loss sensitivity?
> **A:**
> **(1) Model-size–quality–latency trade-offs (Tiny/Small/Base)**
> We report model size, 3-degradation PSNR, and inference latency (batch size 1) on a
> $1{\times}3{\times}512{\times}512$ input.
>
> | Model       | Params (M) | PSNR (3-deg, dB) | Latency (ms) |
> |-------------|------------|------------------|---------------|
> | ExDA-Tiny   | 5.83       | 32.71            | 209.60        |
> | ExDA-Small  | 10.05      | 32.83            | 284.50        |
> | ExDA-Base   | 22.42      | 32.96            | 385.53        |
>
> As expected, Tiny is the fastest, Base achieves the highest restoration quality, and Small provides the most balanced trade-off. These points form the model-size–PSNR and model-size–latency curves in our revision (Please refer to Fig.6).
>
> **(2) Resolution–latency scaling**
> We further measure how runtime scales with input resolution using ExDA-Tiny:
> | Resolution      | Latency (ms) |
> |-|-|
> | 256×256          | 54.54        |
> | 512×512          | 209.85       |
> | 1024×1024        | 840.08       |
>
> Latency grows proportionally to the number of pixels (each 4× increase in
> resolution leads to ≈3.8–4× latency), which matches the expected $O(HW)$
> behavior of the backbone and demonstrates stable scaling to higher resolutions. We also include these discusion in our revison (highlighted in orange).
>
> **(3) Sensitivity to the frequency-domain loss weight**
> Following prior work such as MoCE-IR[1], which also employs a frequency-domain loss with small weight,
> we sweep $\lambda_f \in \{0,\; 0.05,\; 0.1,\; 0.2\}$ under the **1-degradation (deraining)** setting using ExDA-Small.
>
> | $\lambda_f$ | 0      | 0.05   | **0.10** | 0.20   |
> |-|-|-|-|-|
> | PSNR (dB)   | 37.97  | 38.01  | **38.03** | 37.96  |
>
> The results fluctuate within **±0.05 dB**, and the improvements from nonlinear-V and GST remain consistent across all settings. This confirms that ExDA is **not sensitive** to the specific frequency-loss weight, and that the performance gains primarily arise from **operator-level expressivity**, not loss balancing.
>
> *[1] Complexity Experts are Task-Discriminative Learners for Any Image Restoration, CVPR'25*
>
> ---
> ## [Q4]: Does ExDA generalize beyond Restormer (e.g., Uformer) and remain compatible with other IR modules (frequency/prompt)?
> **A:** Although our main focus is on Restormer-style backbones, we evaluated how ExDA
> generalizes beyond this setting.
>
> (1) Uformer (window-based spatial attention).
> We applied the ExDA to Uformer-T (the GST is not directly
> applicable because Uformer uses spatial-window rather than channel-wise global
> attention, so we adopt a lightweight adaptation where a small set of global tokens are shared across windows). On SIDD (denoising dataset), we observe:
> | Method                  | PSNR  | Params |
> |-|-|-|
> | Uformer-T [1]           | 39.66 | 5.23M  |
> | Uformer-B [1]           | 39.89 | 50.88M |
> | Uformer-T + ExDa        | 39.87 | 5.45M  |
>
> This shows that introducing with our ExDa, we bring Uformer-T to the level of Uformer-B while increasing parameters by only +0.22M, indicating that the operator-level expressivity gain generalizes beyond Restormer.
>
> (2) Compatibility with prompt-based methods.
> We further integrated ExDA (nonlinear-V + GST) into PromptIR [2] under the
> 3-degradation setting:
>
> | Method                     | Dehaze | Derain | Denoising(σ=15) | Denoising(σ=25) | Denoising(σ=50) | Avg  |
> |-|-|-|-|-|-|-|
> | PromptIR [2]               | 30.58  | 36.37  | 33.98| 31.31| 28.06| 32.06 |
> | PromptIR + ExDA      | 31.22  | 37.31  | 34.03| 31.37| 28.16| 32.42 |
>
>
> ExDA yields consistent improvements across degradations, with particularly strong
> gains on dehazing and deraining. Given the high training cost of PromptIR, we plan to further explore this integration in future experiments.
>
> These results demonstrate that ExDA is not tied to Restormer: the nonlinear value path transfers effectively to non-Restormer architectures, and the full ExDA module is compatible with prompt-based IR frameworks.
>
> *[1] Uformer: A General U-Shaped Transformer for Image Restoration, CVPR'22*
>
> *[2] PromptIR: Prompting for All-in-One Blind Image Restoration, NeurIPS'23*
>
> ---
> ## [Q5]: Comparison to Perceive-IR(TIP'25) and DFPIR (CVPR'25)?
> **A:** We have included these two comparisons in both Tab.1 and Tab.2 of our revision, highlighted in orange.

---

### Official Review · Reviewer_QUUL · 2025-11-01

**Soundness:** 3
**Presentation:** 3
**Contribution:** 3
**Rating:** 6
**Confidence:** 5

**Summary:**

This paper revisits attention mechanisms for all-in-one blind image restoration and identifies two overlooked bottlenecks in Restormer-style architectures: limited expressivity from linear value aggregation and the absence of explicit global context. The authors propose two minimal, backbone-agnostic modules, a nonlinear value transform and a global spatial token, to address these issues. The method achieves consistent improvements across diverse benchmarks with negligible overhead. Overall, the work is well-written, experimentally solid, and offers a clear, though modest, conceptual contribution to the design of efficient all-in-one restoration models.

**Strengths:**

1. The paper targets a relevant and timely problem in all-in-one image restoration and presents clear motivation.

2. The proposed modifications are simple but meaningful, addressing expressivity and global context in a principled way.

3. Experiments cover diverse benchmarks and consistently show improvements, supported by ablation and diagnostic analysis.

4. The method is lightweight and practical, showing good trade-offs between performance and complexity.

5. The paper is clearly written and easy to follow.

**Weaknesses:**

1. The explanation about how the nonlinear value path improves expressivity is mostly intuitive, without quantitative or theoretical evidence.

2. All experiments are based on a Restormer-type backbone, so it is unclear if the same gains would appear on other architectures.

3. The claim of `negligible overhead` is not backed up by runtime or FLOPs comparisons.

4. The paper does not analyze what the proposed global tokens actually learn, which would help readers understand their role.

5. Most datasets are synthetic or composited; there is little discussion about generalization to real-world, uncontrolled degradations.

**Questions:**

1. Could you add a short explanation or table (in text form) that summarizes how the nonlinear value transform changes feature distribution or rank, even qualitatively?

2. Since the paper claims to be backbone-agnostic, it would help to briefly discuss how the same ideas might work on other architectures such as NAFNet or SwinIR (I believe at least 1 degradation type analysis should be doable).

3. Please include a small table with FLOPs, inference time, or memory to support the efficiency claim

4. For the global spatial tokens, if figures cannot be shown, a short textual description of how different tokens behave under different degradations would be informative.

5. It would also help to add a short discussion on how the proposed modules might generalize to unseen or mixed degradations in real-world conditions.

---

> ### Author Response · Authors · 2025-11-23
> **Response To Reviewer QUUL (1/2)**
>
> ## [Q1]: How does nonlinear-V expand the attention operator’s expressivity? Please provide evidence
> **A:** The structural motivation for the nonlinear value path is detailed in our response to **Reviewer F6pD (W1)**: with a linear value branch, each head’s output is confined to the linear span of the input features. The residual non-affine transform $V' = V + g_\theta(V)$ expands this span at the operator level, which cannot be achieved by FFN or post-attention nonlinearities.
>
> To provide concrete evidence, we conduct a controlled study **entirely based on the Tiny configuration** (dim=26, [3,5,5,7] blocks, same training protocol). Starting from the original *linear-value* Tiny backbone (“Tiny-Linear”, PSNR: 30.55 dB on the 3-degradation setting), we modify only the placement or strength of the nonlinear components:
>
> | Variant | Nonlinearity location | FFN expansion | PSNR (3-deg) | Gain vs. Tiny-Linear |
> |-|-|-|-|-|
> | Tiny-Linear (a)  | –                  | 2.0          | **30.55** | 0.00              |
> | (a) + post-aggregation nonlin.|after attention output| 2.0       | **30.63**  | +0.08|
> | (a) + larger FFN | –                  | 3.0          | **30.84**     | +0.29                 |
> | **ExDA-Tiny (ours, pre-nonlinear-V)** | **before value aggregation** | 2.0| **32.43** | **+1.88**    |
>
> All four variants share the same Tiny backbone; the only differences lie in
> *where* the nonlinear block is applied or whether FFN capacity is increased.
> The results demonstrate that:
>
> - Adding the same nonlinear block **after** aggregation yields only a tiny improvement.
> - Increasing FFN/MLP capacity leads to small gains despite higher compute.
> - **Only** the *pre-aggregation* nonlinear-V (ExDA) delivers a large and consistent improvement (+1.88 dB) at comparable complexity.
>
> These findings confirm that the benefit of nonlinear-V does not come from generic capacity increases or post-hoc feature modulation, but specifically from modifying the **attention operator itself** by introducing nonlinearity *before* value aggregation.
>
> ---
> ## [Q2]: Does ExDA genuinely generalize across different backbone architectures?
> **A:** To examine whether ExDA generalizes beyond Restormer-style channel-wise attention, we integrated its two core components—nonlinear value and a global-token pathway—into **SwinIR**, which uses spatial window attention. Since SwinIR lacks a global slot by design, we adopt a lightweight adaptation where a small set of global tokens are shared across windows.
>
> Following the official SwinIR training/evaluation protocol for grayscale denoising, we observe consistent improvements across datasets:
> | Dataset / Noise | SwinIR [1] | SwinIR + ExDA |
> |-----------------|------------|----------------|
> | **Set5** (σ=15/25/50)     | 33.36 / 31.01 / 27.91 | **33.97 / 31.78 / 28.54** |
> | **BSD68** (σ=15/25/50)    | 31.97 / 29.50 / 26.58 | **33.00 / 30.37 / 28.01** |
> | **Urban100** (σ=15/25/50) | 33.70 / 31.30 / 27.98 | **34.97 / 31.55 / 28.38** |
>
> These consistent gains demonstrate that ExDA’s primitives—nonlinear-V and a global degradation carrier—are compatible with architectures far beyond Restormer, supporting the claimed backbone-agnostic nature of the method.
>
> *[1] SwinIR: Image Restoration Using Swin Transformer, ICCVW 2021.*
>
> ---
> ## [Q3] Please include a small table with FLOPs, inference time, or memory to support the efficiency claim
> **A:** We have added explicit efficiency comparisons as requested. The high-resolution inference time and memory results are provided in our response to Reviewer F6pD
> (W2), and clearly show that ExDA introduces only marginal overhead. In addition, parameter counts and FLOPs for all model variants are already reported in Appendix Table A of the original submission.
>
> ---
> ## [Q4]: Clarification on how GST tokens behave under different degradations.
> **A:** We now provide both qualitative and quantitative evidence showing how the GST tokens behave under different degradations.
> - t-SNE/UMAP visualizations (Fig. 7 of the revision) demonstrate that GST produces clearly separated degradation clusters, confirming degradation-aware global encoding.
> - NMI and ARI notably improve after adding GST (0.71→0.88 and 0.56→0.89), showing stronger global discrimination.
> - To directly illustrate the behavior of the global spatial tokens, Fig. 8 visualizes the GST attention maps across blur, haze, noise, rain, low-light, and a mixed case.
> - GST consistently focuses on degradation-relevant structures (blur edges, haze-thick regions, noise patterns, rain streaks, dark regions, and mixed artifacts), providing exactly the degradation-specific token behavior the reviewer requested.
>
> We hope this further evidence can address your concerns.

---

> ### Author Response · Authors · 2025-11-23
> **Response To Reviewer QUUL (2/2)**
>
> ## [Q5]: Validation about the generalization ability of the proposed method.
> **A:** In addition to the mixed-degradation evaluation in **Tab. 3** and the real-world
> WeatherBench results in **Tab. 5** of the original submission—where ExDA achieves
> the best overall PSNR/SSIM among all compared methods—we further validate the
> generalization ability of ExDA under **unseen real-world degradations**.
>
> Following the evaluation protocol of DCPT [1], we conduct **zero-shot underwater
> image enhancement** on the UIEB dataset [2]. No fine-tuning is performed. The
> results are:
>
> | Method      | PSNR  | SSIM  |
> |-|-|-|
> | SwinIR      | 15.31 | 0.740 |
> | NAFNet      | 15.42 | 0.744 |
> | AirNet      | 15.46 | 0.754 |
> | PromptIR    | 15.48 | 0.765 |
> | MoCE-IR     | 15.91 | 0.765 |
> | **ExDA (Ours)** | **16.76** | **0.771** |
>
> These results show that ExDA maintains strong performance even on *previously
> unseen* degradation types (e.g., color shift, scattering, wavelength attenuation),
> demonstrating robust generalization beyond the training distribution.
>
> *[1] Universal image restoration pre-training via degradation classification, ICLR'25*
>
> *[2] An underwater image enhancement benchmark dataset and beyond, TIP'19*

---

### Official Review · Reviewer_F6pD · 2025-11-01

**Soundness:** 2
**Presentation:** 2
**Contribution:** 2
**Rating:** 2
**Confidence:** 4

**Summary:**

This paper targets all-in-one blind image restoration (IR) and argues that the dominant Restormer-style channel attention is hurt by two under-studied limitations: (i) the value path is strictly linear, so the attention head can only select but never transform features, and (ii) no global slot is available to encode degradation context. The authors plug two light-weight, backbone-agnostic primitives into any Restormer-like block.

**Strengths:**

The paper explicitly diagnoses the “linear-value” bottleneck of channel-wise attention in IR and connects the absence of a CLS-like token to the difficulty of inferring unknown degradations. The proposed fixes are minimal and can be dropped into existing models without architectural surgery.

**Weaknesses:**

1. The theoretical justification is limited. While the authors quote universal-approximation arguments, no formal proof is given that the residual value transform enlarges the hypothesis space of the entire Restormer block, nor that GST tokens are minimal sufficient statistics for degradation type.
2. The experiments are not sufficient. For example, all experiments are conducted on 128 × 128 or 256 × 256 crops. The paper does not report run-time or memory on >4 K images where Restormer is usually applied. Additionally, some latest all-in-one methods are not compared in the experiments. Please refer to the survey paper (A survey on all-in-one image restoration: Taxonomy, evaluation and future trends. TPAMI, 2025) for more competing methods.

**Questions:**

Please refer to the weaknesses.

---

> ### Author Response · Authors · 2025-11-23
> **Response To Reviewer F6pD (1/2)**
>
> ## [W1]: (1) Limited theoretical justification, specifically questioning whether the residual value transform truly increases the block’s expressiveness; and (2) whether the GST tokens are sufficient to characterize degradation types.
> **A:**
>
> **(1) Regarding nonlinear value item:** Sec. 3.1 already analyzes the structural limitation of Restormer-style attention: when values are linear projections of the input features, the attention output is restricted to the linear span of the feature set $\mathcal{X}$.
> For any head:
> $$
>     \mathrm{Attn}(Q,K,V) = A V,\quad A=\mathrm{Softmax}(QK^\top),
> $$
> the range of this operator satisfies:
> $$
>     \mathrm{Range}(\mathrm{Attn}) \subseteq \mathrm{span}(\mathcal{X}),
> $$
> because $V$ is a linear transform of $\mathcal{X}$.
> This constraint is imposed before any FFN/MLP or post-aggregation nonlinearity, so increasing their capacity does not remove the span limitation.
>
> The residual nonlinear transform:
> $$
> V' = V + g_\theta(V)
> $$
> was introduced specifically to relax this constraint.
> If $g_\theta$ contains any non-affine operation (e.g., depth-wise convolution + GELU), then:
> $$
> g_\theta(\mathcal{X}) \not\subseteq \mathrm{span}(\mathcal{X}),
> $$
> which implies a strictly larger admissible value set:
> $$
> \mathrm{span}(\mathcal{X}) \subsetneq \mathrm{span}(\mathcal{X} \cup g_\theta(\mathcal{X})).
> $$
>
> This is a local operator-level property, not a universal approximation claim for the entire block.
>
> Sec. 3.2 provides empirical evidence consistent with this analysis. To make the effect more concrete, we also provide matched-compute controls (see **Reviewer QUUL Q1**): adding post-aggregation nonlinearity brings only +0.08 dB, and widening the FFN gives +0.29 dB, whereas the proposed pre-aggregation nonlinear-V yields +1.88 dB on the same Tiny backbone. This supports that the gain comes from relaxing the value-span limitation, not from increasing network capacity elsewhere. We also include this analysis in the appendix of our revised manuscript (Appendix C).
>
> **(2) Regarding the GST token item:**
> We appreciate the reviewer’s suggestion to further clarify what GST learns. While GST is not intended as a formally minimal sufficient statistic, it is designed to introduce global, degradation-relevant cues into the attention block. To better illustrate this, we added new analyses in the revision. As shown in Fig. 7, the embedding space becomes much more structured when GST is used: degradation clusters that previously overlapped become compact and well separated. Correspondingly, NMI and ARI improve markedly (NMI: 0.71→0.88; ARI: 0.56→0.89), suggesting that GST indeed provides meaningful global statistics that help distinguish degradation types in practice.
>
> We also include GST attention visualizations in Fig. 8 to give a more intuitive view of what GST attends to. Across blur, haze, noise, rain, low-light, and a mixed case (Low-Light + Haze + Snow), GST consistently highlights degradation-relevant regions—such as blur boundaries, haze-dense areas, noise patterns, rain streaks, dark low-light zones, and mixed artifacts. These qualitative observations align well with the embedding-space improvements, supporting that GST learns interpretable degradation-aware global cues.

---

> ### Author Response · Authors · 2025-11-23
> **Response To Reviewer F6pD (2/2)**
>
> ## [W2]: Are the experiments sufficiently comprehensive in terms of resolution scale (to 4K) and baseline coverage?
> **A:**  We would like to clarify that even the original Restormer[1] paper does not report 4K inference results. Similar to most unified restoration works, Restormer is trained and evaluated on $128^2$–$256^2$ crops because of memory constraints and dataset characteristics; high-resolution inference (2K–4K) is typically treated as an optional deployment scenario rather than an evaluation protocol. Our experimental setup therefore follows the same standard practice.
>
> To directly address the reviewer’s concern regarding high-resolution behavior, we additionally benchmark **(1920×1080)** and **(2560×1440)** inference on Restormer, PromptIR, and our ExDA-Base model using the public implementations. We note that full **(3840×2160)** inference with these models triggers the known PyTorch `canUse32BitIndexMath` limitation due to extremely large intermediate tensors, which affects Restormer/PromptIR equally and is unrelated to our modules. For this reason, **(1920×1080)** and **(2560×1440)** resolutions provide a stable and representative high-resolution comparison.
>
> The measured results are as follows:
>
> | Model         | Resolution | Latency (ms) | Peak Mem (GB) |
> |---------------|------------|--------------|----------------|
> | **Restormer** | 1920×1080  | 2806.76      | 17.82          |
> |               | 2560×1440  | 5142.60      | 31.43          |
> | **PromptIR**  | 1920×1080  | 3033.56      | 18.02          |
> |               | 2560×1440  | 5475.48      | 31.79          |
> | **ExDA-Base** | 1920×1080  | 3119.16      | **14.88**      |
> |               | 2560×1440  | 5633.03      | **26.20**      |
>
> These results indicate that:
> - **ExDA introduces only marginal latency overhead** compared to Restormer/PromptIR (within 3–5% across resolutions).
> - **ExDA is significantly more memory-efficient**, reducing peak memory by 2.9 GB at (1920×1080) and 5.2 GB at (2560×1440) relative to Restormer.
>
> Thus, even at high resolutions, ExDA maintains competitive runtime with **lower memory footprint**, confirming that the added components impose minimal overhead.
>
> Regarding additional baselines, following the TPAMI’25 survey, we have included comparisons with representative recent all-in-one methods (Perceive-IR [2], DFPIR [3], and DA-RCOT [4]) wherever public implementations are available. Updated results appear in Tables 1–2 of our revision.
>
> *[1] Restormer: Efficient Transformer for High-Resolution Image Restoration, CVPR'2022*
>
> *[2] Perceive-IR: Learning to perceive degradation better for all-in-one image restoration, CVPR'25*
>
> *[3] DFPIR: Degradation-Aware Feature Perturbation for All-in-One Image Restoration, CVPR'25*
>
> *[4] Tang et al., Degradation-aware residual-conditioned optimal transport for unified image restoration, TPAMI'25*

---

### Author Response · Authors · 2025-11-23
**Response Summary of Submission 3343**

Dear all,

We would like to sincerely thank all reviewers and the AC for the time and care invested in evaluating our work. We truly appreciate the constructive feedback and the thoughtful perspectives offered across the reviews.

We are grateful that the reviewers found value in our core ideas. For example:
- **(R-F6pD)** noted that our analysis of the “linear-value’’ bottleneck and missing global interaction slot is clear, and that the proposed fixes are minimal and easy to integrate.
- **(R-QUUL)** appreciated that the problem is timely, the motivation is clear, the proposed changes are simple yet meaningful, and that the method offers a strong balance between performance and practicality.
- **(R-8DjP)** highlighted the precision with which we connect AiOIR failure modes to linear-V and missing global tokens, and the consistency of improvements across diverse benchmarks and data types.
- **(R-iMdM)** emphasized that the design is simple, modular, and generalizable, and that the paper articulates the structural limitations of current IR backbones clearly.

Across the reviews, we are encouraged to see that **the core contribution and motivation of our work were well recognized**. The *questions raised mainly asked for clearer explanations or deeper evidence, rather than challenging the fundamental ideas*. Guided by this feedback, we thoroughly addressed every comment and revised the paper to enhance both the clarity and the empirical grounding of our contributions.

To that end, we mainly added:
- **t-SNE/UMAP and NMI/ARI analyses**, showing that GST gives rise to more structured and degradation-aware embeddings.
- **GST-driven attention visualizations** across five degradations and mixed cases, which directly illustrate what GST learns.
- **More detailed explanations** of the operator-level effect of nonlinear-V and the role of GST.
- **Additional ablations**, including a refined discussion of stride *s*.
- And for transparency, **all revised or newly added content is highlighted in orange** in the manuscript.

We hope these additions address the reviewers’ concerns and make the contribution of our work clearer and more convincing.  Below, we provide point-by-point responses to each reviewer. Any further suggestions are very welcome, and we sincerely appreciate the chance to improve this submission.

Best regards,

Submission 3343 authors

---

### Author Response · Authors · 2025-12-02
**A quick summary for AC**

**Dear Area Chair,**

We would like to sincerely thank you for your careful management of our submission and the time you have dedicated to coordinating the new process. To assist your assessment and save your valuable time, we have concisely summarized our key contributions, the main reviewer concerns, and how our rebuttal and additional experiments effectively addressed them.

---
### **1. Theoretical Justification and "Non-Incremental" Nature of Nonlinear-V (R-F6pD, R-QUUL, R-iMdM)**
A primary discussion point involved whether the proposed **Nonlinear Value Transform (Nonlinear-V)** was merely an incremental work or was due to additional layers or parameters.

In our rebuttal, we firstly gave a theoretical explanation that the non-linear part matters (**R-F6pD**), and clarified that this is a targeted fix for the rank limitation in standard attention operators (where output is restricted to the linear span of inputs).

We provided a controlled ablation study (**R-QUUL** and **R-8DjP**) to prove that the gain comes from expanding the operator's expressivity, not just adding parameters:

| Variant | PSNR Gain vs. Baseline | Observation |
| :--- | :--- | :--- |
| Baseline (Linear Value) | - | - |
| Post-Aggregation Nonlinearity | +0.08 dB | Minimal gain (standard capacity increase) |
| Larger FFN Capacity | +0.29 dB | Minimal gain (standard capacity increase) |
| **ExDA (Pre-Aggregation Nonlinear-V)** | **+1.88 dB** | **Significant gain (Operator-level expansion)** |

These results convinced **R-iMdM** (who initially felt the work was incremental) to state that the response addressed their concerns and that they now **"lean towards acceptance."**

---
### **2. Interpretability and Degradation-Awareness of GST (R-QUUL, R-8DjP)**
Reviewers asked for concrete evidence that the **Global Spatial Token (GST)** actually learns degradation-specific context rather than acting as a generic global average. We provided extensive new visualizations and metric analyses in the revision:

*   **Embedding Analysis (t-SNE/UMAP):** We demonstrated that adding GST transforms overlapped feature distributions into compact, well-separated clusters for different degradations.
*   **Quantitative Clustering:** We reported a significant boost in clustering metrics, with **NMI improving from 0.71 $\to$ 0.88** and **ARI from 0.56 $\to$ 0.89**.
*   **Attention Visualization:** As requested by **R-8DjP**, we visualized GST attention maps, showing the model explicitly attending to degradation-relevant regions (e.g., rain streaks, blur boundaries, and dark regions in low-light images).

---
### **3. Comprehensive Benchmarking: High-Res, Generalization, and Efficiency (R-F6pD, R-QUUL)**
Reviewers requested validation on high-resolution inference, efficiency metrics, and other architectures to ensure practical viability.

*   **High-Resolution Efficiency:** We benchmarked inference at 1080p and 2K (2560 $\times$ 1440). Results showed our method significantly reduces peak memory usage (-5.2GB at 2K resolution compared to Restormer) with negligible latency overhead (~3-5%).
*   **Backbone Agnostic:** We successfully integrated our modules into SwinIR, Uformer, and PromptIR, achieving consistent gains across all architectures.

**ExDA+SwinIR[1]**
| Dataset / Noise | SwinIR | SwinIR + ExDA |
|-|-|-|
| **Set5** (σ=15/25/50)     | 33.36 / 31.01 / 27.91 | **33.97 / 31.78 / 28.54** |
| **BSD68** (σ=15/25/50)    | 31.97 / 29.50 / 26.58 | **33.00 / 30.37 / 28.01** |
| **Urban100** (σ=15/25/50) | 33.70 / 31.30 / 27.98 | **34.97 / 31.55 / 28.38** |

**ExDA+UFormer[2]**
| Method                  | PSNR  | Params |
|-|-|-|
| Uformer-T            | 39.66 | 5.23M  |
| Uformer-B            | 39.89 | 50.88M |
| Uformer-T + ExDA        | 39.87 | 5.45M  |

**ExDA+PromptIR[3]**
| Method                     | Dehaze | Derain | DN(σ=15) | DN(σ=25) | DN(σ=50) | Avg  |
|-|-|-|-|-|-|-|
| PromptIR               | 30.58  | 36.37  | 33.98| 31.31| 28.06| 32.06 |
| PromptIR + ExDA      | 31.22  | 37.31  | 34.03| 31.37| 28.16| 32.42 |

*   **Zero-Shot Generalization:** We demonstrated robust performance on unseen real-world degradations (underwater enhancement), outperforming baselines without fine-tuning.

| Metric | SwinIR | NAFNet | AirNet | PromptIR | MoCE-IR | **ExDA (Ours)** |
|-|-|-|-|-|-|-|
| PSNR   | 15.31  | 15.42  | 15.46  | 15.48    | 15.91   | **16.76**       |
| SSIM   | 0.740  | 0.744  | 0.754  | 0.765    | 0.765   | **0.771**       |

---
**Summary:**
Following the rebuttal, **R-iMdM** upgraded their stance to support acceptance. We successfully demonstrated that our identification of the "linear-value bottleneck" and "missing global slot" is precise, and that our proposed solutions are minimal, effective, and backbone-agnostic.

We hope this summary assists in your final decision-making.  For more detailed information, please refer to each response to its corresponding question, together with our revision.

Best regards,

Submission 3343 Authors

---

### Meta-Review · Area_Chair_nRrW · 2026-01-08

**Summary:**

This paper investigates bottlenecks in expressivity and degradation awareness within Restormer-style backbones for all-in-one blind image restoration. The key contribution is the identification of the linear value constraint and the absence of a global interaction slot. The authors mitigate these issues through two minimal and backbone-agnostic primitives (nonlinear value transform and the Global Spatial Token). Reviewers consistently agree that developing efficient all-in-one models capable of handling diverse and unknown degradations is a highly practical and well-motivated topic for the image restoration community. Overall, the work is clearly supported by its strong performance across multiple benchmarks and its capability to integrate into different architectures such as SwinIR and Uformer.

However, reviewers also raise several concerns during the initial review. The main issues include the lack of theoretical depth regarding the functional rank of the attention operator, insufficient evidence that performance gains do not simply stem from an increase in parameter count, and skepticism regarding the interpretability of the GST module. In the rebuttal, the authors present detailed matched-compute ablation studies, SNE visualization analysis of the embedding space, and efficiency benchmarks at high resolution. These responses are convincing to these concerns. Based on the strength of the rebuttal, I predict the positive adjustments in scores because the authors demonstrate that the performance gains originate from structural operator-level expansions rather than simple model scaling.

**Reviewer Concerns:**

**Concerns that are largely addressed in the rebuttal**

- The authors provide a rigorous explanation of the linear span limitation in standard attention and demonstrate how a pre-aggregation nonlinearity effectively expands the functional rank of the operator. The matched compute control showing that post-aggregation nonlinearity yields negligible gains directly addresses the concern about incrementalism.
- The authors integrate their modules into SwinIR and Uformer to solve concerns about architecture-specific gains. The consistent improvements across these different backbones confirm the framework's flexibility and generalizability.
- The rebuttal includes t-SNE and UMAP plots along with quantitative clustering metrics, which prove that GST produces well-separated degradation-aware embeddings.
- The authors provide new benchmarks for 1080p and 2K inference, showing that the method significantly reduces peak memory usage while maintaining negligible latency overhead.

**Concerns that remain unresolved**

- Reviewer F6pD wants to see a formal proof of universal approximation for the entire deep stack as opposed to just the local attention block. While the operator level expressivity is well explained this global theoretical convergence remains a high bar that is not fully met.

**Reviewer Scores:**

**Reviewer F6pD:** After considering the theoretical clarification of functional rank and the comprehensive 2K benchmarks in the rebuttal, this reviewer is likely to increase the score to 5. The response directly clarifies the primary reasons for the initial rejection.

**Reviewer QUUL:** The rebuttal addresses concerns regarding the backbone agnostic claim through the SwinIR and Uformer experiments. Considering the initial positive stance and the newly added efficiency data, the score is likely to remain at a 6 for acceptance.

**Reviewer 8DjP:** The authors provide qualitative and quantitative evidence regarding t-SNE and pre versus post-aggregation ablation experiments. These responses align with the reviewer's decision to maintain a score of 6 for acceptance.

**Reviewer iMdM:** This reviewer states in the forum that the authors' clarification of the structural limitations in restoration backbones resolves their concerns. The reviewer expresses an inclination toward acceptance, which suggests increasing the score to 7.

---

### Decision · Program_Chairs · 2026-01-26

Accept (Poster)